# Local structure-function relationships in human brain networks across the lifespan

Farnaz Zamani Esfahlani[1], Joshua Faskowitz [1,2], Jonah Slack[1], Bratislav Mišić [3] & Richard F. Betzel [1,2,4,5 ✉]

A growing number of studies have used stylized network models of communication to predict brain function from structure. Most have focused on a small set of models applied globally. Here, we compare a large number of models at both global and regional levels. We find that globally most predictors perform poorly. At the regional level, performance improves but heterogeneously, both in terms of variance explained and the optimal model. Next, we expose synergies among predictors by using pairs to jointly predict FC. Finally, we assess age-related differences in global and regional coupling across the human lifespan. We find global decreases in the magnitude of structure-function coupling with age. We find that these decreases are driven by reduced coupling in sensorimotor regions, while higher-order cognitive systems preserve local coupling with age. Our results describe patterns of structure-function coupling across the cortex and how this may change with age.

[1] Department of Psychological and Brain Sciences, Indiana University, Bloomington, IN 47405, USA. [2] Program in Neuroscience, Indiana University, Bloomington, IN 47405, USA. [3] McConnell Brain Imaging Centre, Montréal Neurological Institute, McGill University, Montréal, QC, Canada. [4] Cognitive Science Program, Indiana University, Bloomington, IN 47405, USA. [5] Network Science Institute, Indiana University, Bloomington, IN 47405, USA.
✉email: rbetzel@indiana.edu

The human connectome constitutes the complete set of neural elements and their anatomical connections to one another[1]. At the macroscale, the connectome can be represented as a graph or network whose nodes and edges correspond to brain regions and white-matter fiber tracts[2]. The topological organization of the connectome is non-random and exhibits small-world architecture[3], hubs and rich clubs[2,4], modules[5], and cost-effective spatial embedding[6]. These structural features constrain patterns of inter-areal communication[7–10], inducing correlations in regional activity, i.e. functional connectivity[11,12].

Estimates of connectomes and functional networks can be obtained easily thanks to easy-to-use processing and reconstruction pipelines[13,14]. However, linking the two connectional modalities to one another has proven challenging, leading to many different approaches[15,16]. Some of the earliest studies of structure-function coupling constrained generative, biophysical models of brain activity with anatomical connections, noticing that the correlation structure of the synthetic time series output by the model was, itself, correlated with both the anatomical network and empirical functional connectivity[7,8,17,18]. Other studies have opted for a simpler, albeit amechanistic, approach and computed correlations between anatomical and functional connection weights[2,19], sometimes using multivariate methods[20].

Although scientifically profitable, both approaches have limitations. Biophysical models, for instance, entail high computational costs that preclude exhaustive searches of parameter spaces. Conversely, statistical and correlative approaches provide limited insight into the mechanisms that support structure-function coupling. Situated between these two extremes, however, are stylized models of interregional communication[12]. In general, these models are based on simple dynamical processes, e.g. diffusion[11], epidemic spreading[21], shortest-paths routing[22], whose solutions can be expressed analytically, and entail low computational costs. As a result, this approach allows users to flexibly implement and adjudicate between different models of communication[23].

Broadly, the space of communication models can be organized along a single axis, with models positioned according to how "centralized" or "decentralized" their communication policy is[12,22]. Shortest paths, for instance, are an example of a centralized communication policy in that using a shortest path for signaling requires complete knowledge of a network's global topology. That is, a particle (signal) moving from a source region to a target region needs to "know" which nodes are on the shortest path and which connections need to be traversed to stay on that path. In contrast, "decentralized" communication policies require no knowledge and include diffusion processes (random walks)[24] and network navigation[25] - where a particle moves from one node to another according to some greedy policy, e.g. choose the connected neighbor nearest the target in some metric space. Alongside centralized and decentralized models are similarity-[26] and distance-based measures[23], which assess the overlap of brain regions' connectivity patterns or their nearness to one another in space.

Although communication models are becoming increasingly common and have been used recently to study individual differences in phenotypes and traits[27], cognition[28], and for subject identification[29], there remain many open questions and frontiers. First, most studies focus on a select set of communication measures and do not compare the performance of those measures against other models. Second, most studies have focused on using communication models to explain variation in whole-brain functional connectivity. In contrast, several recent studies have shown that structure-function coupling is heterogeneous across the cortex[30,31], suggesting variation in the underlying communication process and motivating further study. Finally, neither of these questions have been addressed in an applied context. Consider, for instance, the human lifespan. Although many studies have independently documented differences in structural[32–34] and functional connections[35,36] through development, maturation, and adulthood, how they evolve jointly in terms of communication models and at the level of brain regions is unknown[28,37].

Here, we address these limitations directly. Using data from the Human Connectome Project[38] and a suite of communication models (predictors) based on both centralized and decentralized processes, topological similarity, and spatial embedding, we show that at the single-subject level, communication models fit at the regional level outperform those fit globally. We also find that predictors based on path length perform poorly, whereas predictors that describe decentralized communication processes perform better. We also show that the variance explained in regional functional connectivity follows a system-specific pattern, with primary sensory systems being more predictable than heteromodal systems. Relatedly, we also find that the distribution of optimal factors vary by brain system. Next, we explore more synergies among predictors, using multi-linear models to predict the weights of functional connections. We find that, among the optimal pairings, a relatively small number of predictors appeared disproportionately often, forming a core set of predictors that, collectively, is essential for predicting regional FC. Finally, we analyze data from the extended Nathan Kline Institute lifespan sample[39]. We show that, globally, FC becomes less explainable across the lifespan, irrespective of best predictor. However, we also show that the prevalence of certain predictors vary stereotypically with age and, although FC predictability decreases with age, the regional pattern of predictability was heterogeneous and largely spared systems associated with executive function and introspection (control and default mode networks). Note that following preprocessing and network construction, we employed a secondary set of criteria for censoring high-motion frames and, if necessary, excluding subjects on the basis of motion. All results described herein were generated using these motion censored data. See Methods for details of exclusion criteria.

## Results

Here, we explore three interrelated questions: Which communication model best explains observed patterns of FC? How does the optimal model vary across cortex? Does the magnitude of coupling and the optimal model vary over the course of the human lifespan? To test these hypotheses, we analyzed two separate structure-function datasets. The first comes from the Human Connectome Project[38] and includes structural and functional connectivity (SC; FC) data from 100 individuals. The second dataset comes from the Nathan Kline Institute-Rockland lifespan sample[39] and includes SC and FC data from 542 individuals. In the following sections, we analyze cortical networks parcellated into $N = 400$ regions of interest[40]. For details concerning data processing and network definition, see Methods.

Our analyses are divided into several sections. In the section Global structure-function coupling is not fully explained by any factor, we investigate individual heterogeneity in terms of which factors best predict whole-brain patterns of FC. Then, in the section Regional structure-function coupling is heterogeneous, we investigate both regional and inter-individual variability in the optimal factor for predicting the FC profiles of single brain regions. Then, in the section Exploiting synergies among predictors leads to increased explanatory power, we use multi-linear models to explain regional patterns of FC. Finally, in the section Structure-function coupling weakens across the human lifespan, we analyze lifespan differences in structure-function relationships as assessed using communication models.

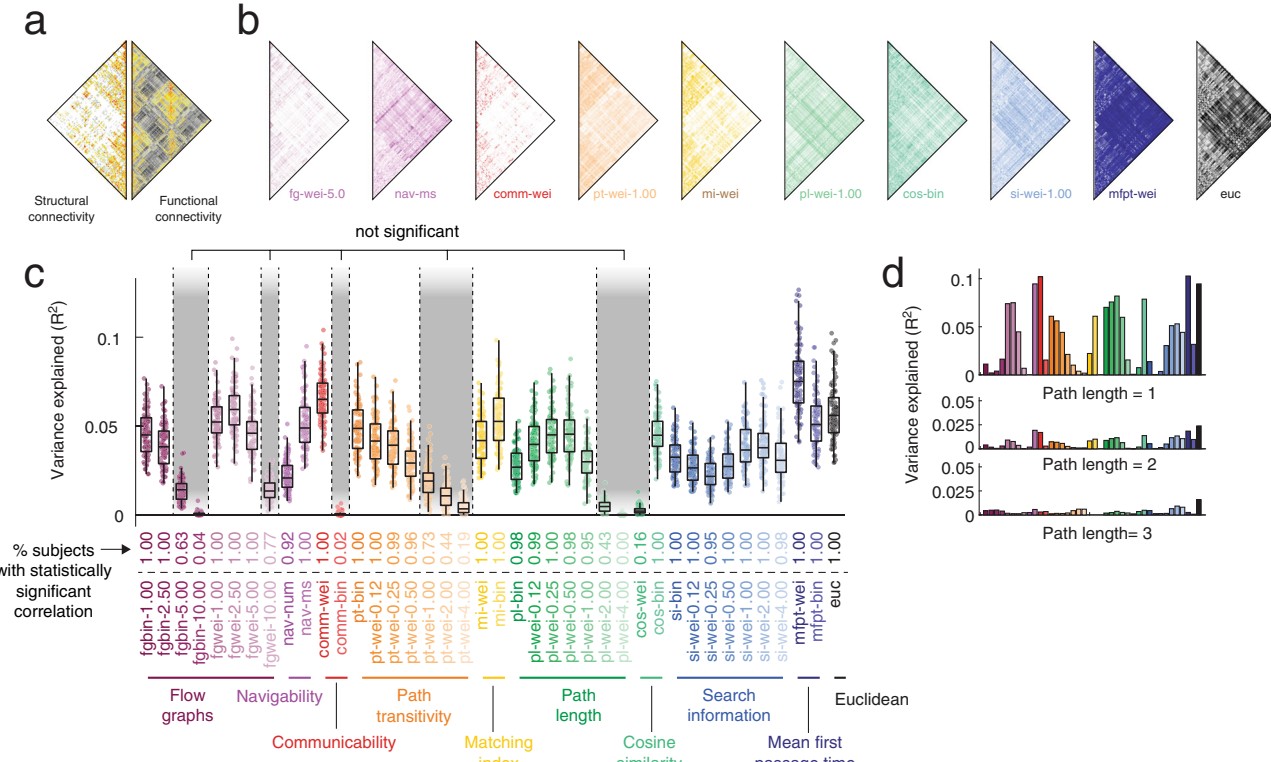

**Fig. 1 Whole-brain functional connectivity is heterogeneous and not fully explained by any factor. a** Whole-brain structural and functional connectivity data from the *HCP* dataset. **b** We used a series of dynamic, topological, and geometric models to transform sparse SC matrices into fully-weighted matrices (predictors). Here, we show examples: flow graphs (*fg-wei-5.0*), navigation (*nav-ms*), communicability (*comm-wei*), path transitivity (*pt-wei-1.00*), matching index (*mi-wei*), path length (*pl-wei-1.00*), cosine similarity (*cos-wei*), search information (*si-wei-1.00*), mean first passage time (*mfpt-wei*), and Euclidean distance (*euc*). Note that here and in all other figures, we adopt a consistent color scheme, wherein data associated with a given predictor, e.g. same measure but different parameters or weighted/binary versions of a measure, are depicted using variation of the same color. For example, communicability measures are red, matching index is yellow, and mean first passage time is dark blue. **c** Variance in whole-brain FC weights explained by factors. Each point represents a subject ($N_{sub} = 70$ subjects following exclusion for motion and data quality; FC and SC averaged over two scan sessions). In each boxplot, the "box" denotes interquartile range (IQR), the horizontal bar indicates the median value, and the whiskers include points that are within 1.5 × IQR of upper and lower bounds of the IQR (25th and 7th percentiles). Any points that fall beyond the whiskers are, by convention, considered outliers. **d** Decreases in variance explained as a function of path lengths (hops). Source data are provided as a Source Data file.

**Global structure-function coupling is not fully explained by any factor.** Recent work has focused on using simple, stylized models to transform sparse SC data into fully-weighted matrices to explain variation in whole-brain patterns of FC (Fig. 1a). In most applications, only a few predictors are investigated, making it difficult to assess the relative performances of different predictors.

Here, we generate a large number of matrices based on SC data from individual subjects. These transformations yield a distinct geometric, topological, or dynamic factor that can then be used to explain variation inter-regional FC. Broadly, we focused on ten classes of predictors: flow graphs parameterized at different timescales[41], two based on greedy navigation[42], two based on communicability[43,44], seven based on path transitivity parameterized at different weight-to-cost transformations[11], two based on the matching index[26], seven based on path length parameterized at different weight-to-cost transformations, two based on cosine similarity, seven based on search information parameterized at different weight-to-cost transformations[45], mean first passage times of random walkers[46], and Euclidean distance. In total, we explored 40 different predictors. In Fig. 1b we show examples of several predictors for a single subject.

First, we assessed whether the FC variance explained was different from one predictor to another. In general, we found high levels of heterogeneity across predictors in terms of their ability to explain the variance in empirical FC (one-way ANOVA;

$F_{(39)} = 270.0$; $p < 10^{-15}$; Fig. 1c). At a single subject level, no predictor explained more than 12.67% of variance. We also assessed the statistical significance of variance explained by each predictor and for each subject, comparing the observed $R^2$ values against those obtained using a spatially-constrained permutation model[47] (1000 permutations). We also identified predictors whose distribution of $R^2$ values across subjects excluded a value of zero (Fig. 1c).

Aggregating across subjects, the best predictors were weighted mean first passage time (*mfpt-wei*; $R^2 = 0.078 \pm 0.019$), weighted communicability (*comm-wei*; $R^2 = 0.066 \pm 0.013$), the flow graph estimated at a Markov time of $t = 2.5$ (*fgwei-2.50*; $R^2 = 0.060 \pm 0.014$), and Euclidean distance (*euc*; $R^2 = 0.057 \pm 0.016$). The remaining factors all explained less variance. Note that these general trends persist, irrespective of whether we examine whole-brain connectivity data or connectivity data based on single hemispheres (see Supplementary Fig. S1).

Across all factors, we found that the majority of variance explained can be attributed to one-step (direct) connections (Fig. 1d). Isolating these connections alone, we found that the average variance explained increased (from 3.4% to 3.9%; paired sample t-test, $p = 8.5 \times 10^{-4}$). However, for multi-step paths, the variance explained decreased substantially.

Collectively, these results suggest that whole-brain FC is not well explained by any single factor in isolation ($\max(R^2) \approx 0.1$)

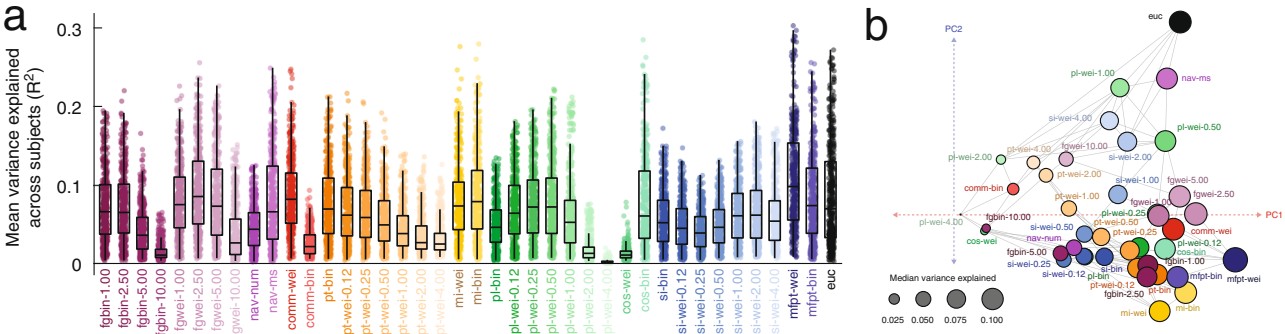

**Fig. 2 Explaining regional FC patterns with dynamic, topological, and geometric factors. a** Distributions of variance explained by different factors. Points represent brain regions and the mean variance explained across the entire cohort. Each point represents a subject ($N_{sub} = 70$ subjects following exclusion for motion and data quality; FC and SC averaged over two scan sessions). In each boxplot, the "box" denotes interquartile range (IQR), the horizontal bar indicates the median value, and the whiskers include points that are within 1.5 × IQR of upper and lower bounds of the IQR (25th and 7th percentiles). Any points that fall beyond the whiskers are, by convention, considered outliers. **b** Spatial embedding of predictors. The coordinates were determined using the following procedure: 1) calculating the similarity of regional correlations for every pair of predictors, 2) thresholding this matrix to retain the $k = 4$ nearest neighbors for each predictor, and 3) performing a principal component analysis of the thresholded and symmetrized matrix. Coordinates represent the first two principal components, PC1 and PC2. Broadly, predictors that are near/distant to one another in principal component space exhibit similar/dissimilar patterns of regional correlations. In this plot, edges exist between predictors if they are nearest neighbors. Note that here the principal component analysis was carried out without z-scoring or centering the columns of the correlation matrix. The size of points is proportional to the mean variance explained across all brain regions. Source data are provided as a Source Data file.

and that the weights of direct connections are more easily explained than indirect connections. Both of these results are in line with previous studies[11] and motivate further investigation into structurally-based explanatory predictors of FC.

**Regional structure-function coupling is heterogeneous.** In the previous section we focused on explaining variance in whole-brain FC using a series of predictors obtained by transforming the SC matrix. In general, we found high levels of heterogeneity and weak correspondence between FC and predictors. In order to achieve better explanatory power, several recent studies have focused on regional patterns of FC and explaining variance in FC from the perspective of individual nodes[30,31,48]. However, these studies were limited in scope to a select set of predictors. Here, we use the same set of 40 predictors to explain regional patterns of FC and identify the optimal factors for each region and brain system.

To explain regional patterns of FC, we fit linear models wherein every row (or equivalently column) in the FC matrix are explained based on the same row in a predictor matrix. Repeating this procedure for every region, predictor, and subject returns a matrix of $R^2$ values whose dimensions are $[400 \times 40 \times 95]$. In general, we find that the overwhelming majority of these models are statistically significant at the single-subject level (see Fig. S2a for results from the HCP dataset). To visualize these results and for subsequent statistical analyses, we averaged over subjects and plotted the mean variance explained for each region and predictor (Fig. 2a). As in the previous section, we found considerable variability across predictors (one-way ANOVA $R^2$; $F(39) = 138.7$; $p < 10^{-15}$) but also across regions (one-way ANOVA $R^2$; $F(399) = 36.0$; $p < 10^{-15}$), confirming that both regions and predictors differ from one another in terms of their mean variance explained. We also compared the spatial similarity between predictors (the similarity of the $400 \times 1$ vectors of variance explained at each region) and used an embedding algorithm to assign predictors locations in two-dimensional space based on that similarity (Fig. 2b).

In general, we found that models seeking to explain local (i.e. regional) variation in FC outperform global models. For each region, we calculated the maximum variance explained by any

model and found values, in some cases, that exceeded 33% variance explained (Fig. 3a). Interestingly, the magnitude of variance explained was, itself, variable across cortex and concentrated within specific sets of brain systems (Fig. 3b, c). In particular, we found that the FC patterns of regions in the somatomotor and visual network were better explained than those of regions in other brain systems (1000 spin test permutations[49]; false-discovery rate fixed at 5%; $p_{adjusted} = 0.036$). Irrespective of brain region, we found that Euclidean distance (*euc*), weighted mean first passage time (*mfpt-wei*), weighted communicability (*comm-wei*), binary cosine similarity (*cos-bin*), and the length of navigation paths in units of Euclidean distance (*nav-ms*), were the most common across subjects, being classified as optimal for 16.8, 15.4, 8.2, 6.3, and 5.0 percent of brain regions. In contrast, the predictors that were least likely to be considered optimal included measures of binary and weighted shortest paths, search information, and flow graphs at long Markov times (See Supplementary Fig. S3 for complete ranking).

Relatedly, we observed that the predictors associated with the maximum variance explained varied across regions and systems (Fig. 3d). We found that within every brain system certain predictors were overrepresented relative to their baseline rate. Consider the visual system, for instance (Fig. 3e). At the population level, 44% of visual regions exhibited FC patterns that were best predicted by their Euclidean distance from other regions. In comparison, the FC patterns of only 12% of control regions were best explained by Euclidean distance (the whole-brain rate is 21%). Interestingly, we found that the control and default mode networks diverged from the whole-brain levels at the highest rate, with 16 and 17 of the 40 predictors overrepresented in these systems, respectively (40% and 42.5%). In contrast, sensorimotor systems (somatomotor and visual) overrepresented only five predictors each (12.5%). These observations align with the putative functional roles of these systems – control and default mode are thought to be polyfunctional while sensorimotor systems subtend a narrower set of functions related to processing specific modalities of information.

Indeed, the predictors were differentially associated with brain regions and systems. To better understand exactly which regions were best explained by a given predictor, we grouped

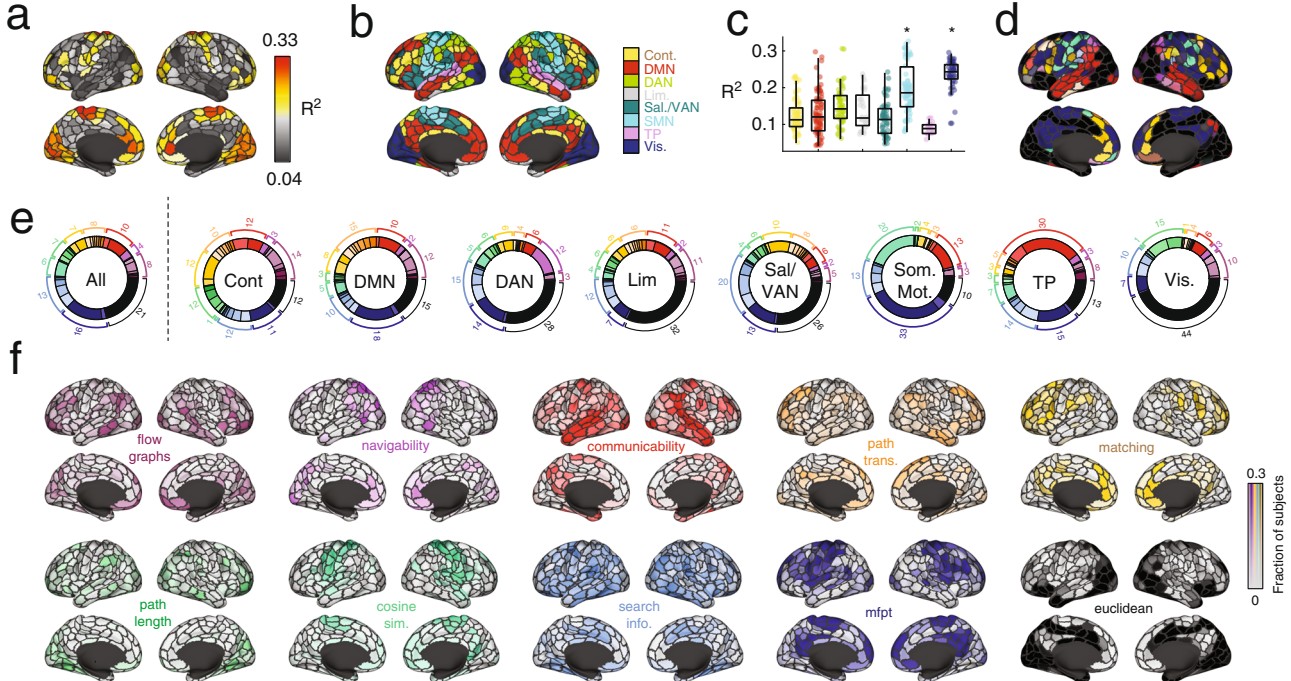

**Fig. 3 Regional and system variation in local coupling patterns. a** The maximum variance explained for each brain region by any of the factors. **b** Brain systems mapped onto cortical surface. **c** Maximum variance explained grouped by brain system. Each point represents a brain region ($N = 400$ parcels defined based on[40]). In each boxplot, the "box" denotes interquartile range (IQR), the horizontal bar indicates the median value, and the whiskers include points that are within $1.5 \times$ IQR of upper and lower bounds of the IQR (25th and 7th percentiles). Any points that fall beyond the whiskers are, by convention, considered outliers. **d** Surface projection of the factors that best explain each region's FC pattern. **e** For each subject we calculated the most predictive factor for every brain region. Here, we group these factors at the level of the entire cerebral cortex (*All*) and at the level of individual systems. **f** For each region and factor, we calculated the fraction of subjects for which that factor explained the most variance. Here, we project these values to the cortical surface. Note here that the colors projected onto the surface are continuous; grays correspond to regions where a predictor was optimal for few subjects while brighter colors correspond to regions where a predictor was optimal for many subjects. In all panels the colorscale was capped at a value of 0.3. Note that the base colors for each surface plot correspond to different predictors and not varying levels of consistency across subjects or statistical significance. Source data are provided as a Source Data file.

predictors into ten broad classes and calculated how each region's FC pattern was best explained by each class of predictor across the 95 subjects (Fig. 3). To assess whether these regional patterns of predictor preference were concentrated within distinct brain systems, we averaged their scores by systems[40] and compared these averages with those obtained under a spatially-aware permutation model[47,49] (1000 permutations; false discovery rate fixed at 5% leading to adjusted critical value of $p_{adj} = 0.0053$). For instance, we found flow graphs overrepresented within the control and default mode networks, navigability within the dorsal attention network, communicability in temporoparietal network, path transitivity in default mode, matching index within control network, path length within the visual and control networks, cosine similarity within the somatomotor network, search information within the control network, and mean first passage time within the temporoparietal network. Interestingly, as a predictor Euclidean distance was not overrepresented within any system, suggesting it lacked clear system specificity.

Finally, we calculated for each region the fraction of subjects for whom each of the 40 predictors was optimal. We treated this distribution as a set of features for each region, describing its preference for one predictor or another. Then, we computed the similarity of these feature vectors for every pair of brain regions, resulting in $400 \times 400$ correlation matrix, which we then clustered using modularity maximization. We found evidence of ten consensus communities, four of which were large and were investigated further (Supplementary Fig. S4). In general, each of

these four communities disproportionately represented a single specific predictor. Namely, weighted mean first passage time (cluster 1), Euclidean distance (cluster 2), weighted communicability (cluster 3), and weighted matching index (cluster 4) (Supplementary Fig. S4c).

We also compared the regional variance explained by communication models with structural connectivity weights alone (Fig. 4). Note that while structural connectivity is sparse and static, all of the communication models yield dense, fully-weighted matrices and many realize stylized dynamical processes. We found that, while the two techniques generated similar regional patterns of explained variance (mean similarity of $r = 0.32 \pm 0.09$; Fig. 4a, b), using communication models as predictors generally outperformed structural connectivity (Fig. 4c, d), with the FC of $68.6 \pm 0.06\%$ of regions being better explained by predictors derived from the SC matrices than the SC matrix itself.

Collectively, these results suggest that global models of interregional communication may fail to account for regional preferences in communication patterns. By fitting explanatory models at the level of regions, we can expose these preferences and heterogeneity across the cerebral cortex in terms of regional predictability.

**Exploiting synergies among predictors leads to increased explanatory power.** In the previous two sections, we demonstrated that at the whole-brain level, measures of communication

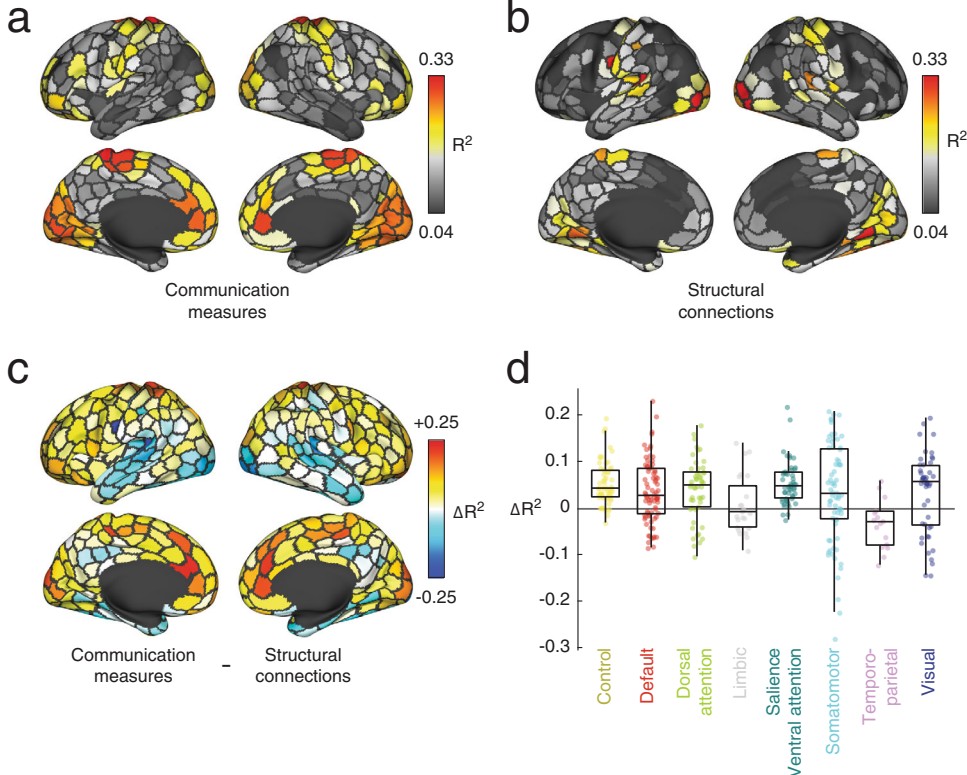

**Fig. 4 Comparison of communication measures and structural connectivity in predicting FC.** In the main text we derived a series of communication measures from sparse structural connectivity (SC) data to predict regional FC patterns. Here, we compare the results from the communication measures with the results obtained from using the structural connections directly. In this analysis, we create a "mask" for each region and subject of its structural connections to other regions. We then extract the weights of those structural and functional connections and fit a linear model to explain the functional connection weights in terms of the structural weights. This results in an $R^2$ value for each region and subject. **a** Variance explained using communication measures averaged across subjects. **b** Variance explained using structural connections alone averaged across subjects. **c** Regional differences in variance explained. Warmer and cooler colors indicate regions whose FC is better predicted using the communication measures and structural connectivity, respectively. **d** Data from panel c but displayed by brain system. Each point represents a brain region ($N = 400$ parcels defined based on[40]). In each boxplot, the "box" denotes interquartile range (IQR), the horizontal bar indicates the median value, and the whiskers include points that are within $1.5 \times$ IQR of upper and lower bounds of the IQR (25th and 7th percentiles). Any points that fall beyond the whiskers are, by convention, considered outliers. Source data are provided as a Source Data file.

explained a small fraction of variance in FC, but could be improved upon by modeling FC at the level of individual brain regions. In both cases, we modeled FC linearly in terms of one predictor and a constant. However, several studies have examined the extent to which combinations of predictors, e.g. search information and Euclidean distance[11,23,31] led to an improvement in variance explained when combined in a multilinear model. Here, we focus on local models while simultaneously building upon those earlier studies to systematically explore all possible combinations of terms.

We constructed linear models of FC based on pairs of predictors. With 40 predictors this yields $40 \times 39/2 = 780$ unique dyads, which we used to explain regional FC variance. To reduce the total number of dyads, we performed the following greedy analysis. For each subject and region, we identified the predictor that explained the greatest fraction of variance in its FC. Then, of the remaining 39 predictors, we identified the one that, when included in the multilinear model, yielded the greatest performance improvement. Then, we counted how frequently specific pairs of predictors appeared together in the multilinear models and summarized the results as a square predictor × predictor matrix (Fig. 5a). Interestingly, we found that the counts followed a heavy tailed distribution, such that a small fraction of predictor pairs appeared disproportionately more frequently than others (Fig. 5b). When we considered the marginal distribution (Fig. 5a,

bar plot at the top of the matrix), we found that Euclidean distance, weighted mean first passage time, weighted communicability, weighted search information (with $\gamma = 4$), and navigability participated in the most dyads, suggesting that these factors, when paired with others, are important for explaining regional patterns of FC.

To better understand the interrelationships among predictors, we modeled the matrix in Fig. 5a as a graph, where nodes and edges correspond to predictors and the frequency with which predictor pairs appear, respectively (see Fig. 5c for an embedding of the network in two-dimensional space). Upon visual examination of the connectivity matrix, it appeared that a small fraction of predictors broadly interacted with others while the remaining predictors weakly interacted with one another. This type of organization is hallmark of core-periphery meso-scale structure, where a densely-connected core of nodes projects to a sparsely connected periphery. In this context, the "core" refers to pairs of predictors (metrics) that frequently appear together in two-term multilinear models. The same core predictors may sometimes be paired with peripheral predictors, but peripheral predictors are infrequently paired together. That is, the core is comprised of predictors that exhibit strong synergies in their ability to predict FC patterns; the periphery is comprised of predictors that exhibit relatively weak synergies. To test whether this type of structure was present, we applied a core-periphery detection algorithm that,

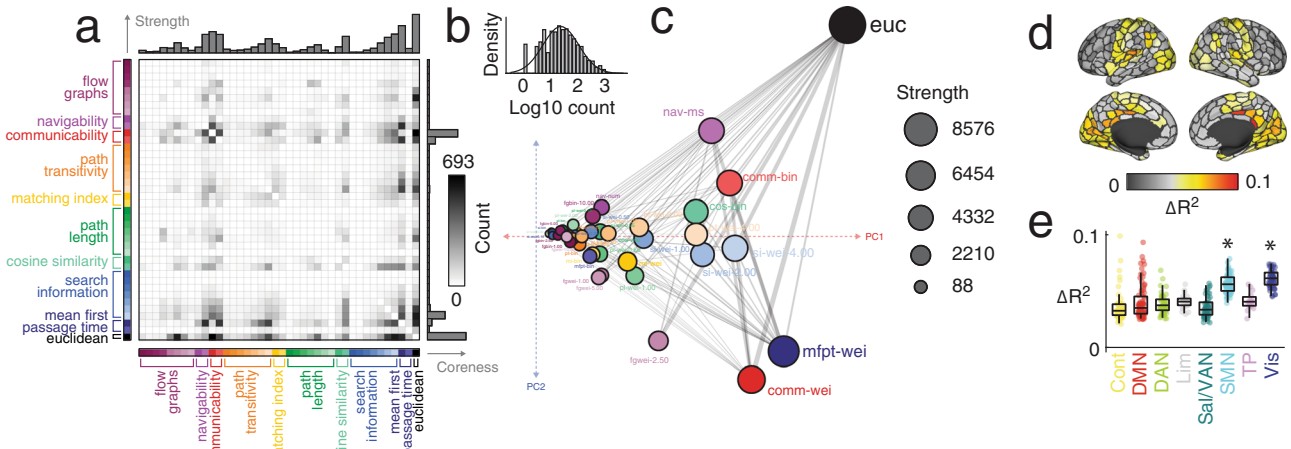

**Fig. 5 Pairwise synergies between predictors.** We used two-predictor multi-linear models to predict regional patterns of FC and identified the optimal pair of predictors for each region and each subject. We then counted how frequently each pair of predictors appeared in the set of optimal predictors. This procedure yields a symmetric matrix of counts (**a**). The counts were approximately log-normally distributed (**b**). The matrix could be modeled as a graph and each node's (predictor's) coreness could be directly calculated (node positions based on principal components analysis of count matrix) (**c**). In this plot, coordinates were determined by: 1) thresholding the count matrix to retain, for each predictor, its $k = 4$ nearest neighbors, and 2) performing a principal component analysis on the thresholded and symmetrized matrix. Here, the coordinates represent the first two principal components, PC1 and PC2. Predictors that are near/distant from one another in principal components space pair with similar/dissimilar sets of predictors when improving $R^2$. Predictors are joined by an edge if they are considered nearest neighbors. As in Fig. 2b, we did not center or z-score columns as part of the principal component analysis. We also calculated the mean regional improvement in $R^2$ from using the multi-linear model *versus* the model with a single predictor. Each point represents a brain region ($N = 400$ parcels defined based on[40]). **d** Improvement ($\Delta R^2$) projected onto the cortical surface. **e** Improvement grouped by canonical brain systems (asterisks indicate statistical significance). In each boxplot, the "box" denotes interquartile range (IQR), the horizontal bar indicates the median value, and the whiskers include points that are within 1.5 × IQR of upper and lower bounds of the IQR (25th and 7th percentiles). Any points that fall beyond the whiskers are, by convention, considered outliers. Source data are provided as a Source Data file.

rather than a binary classification of a node as "core" or "peripheral", assign each node a parameterized and continuous measure of coreness. Here, we systematically varied the two parameters – $\alpha$ and $\beta$ – which control, roughly, the smoothness of the distinction between core and periphery and the number of nodes in the core, respectively. We aggregated coreness scores over the top 5% of parameter pairs to obtain a mean coreness score for every node (see Supplementary Fig. S5). As expected, the predictors with the highest levels of coreness included Euclidean distance, weighted communicability, weighted mean first passage time, and search information (with $\gamma = 4$) (Fig. 5a; right margin). With the exception of Euclidean distance, all of these measures are based on diffusive, decentralized dynamics. Mean first passage time and search information concern random walks over a network, while communicability is associated with the ensemble of multi-step walks through a network. In contrast, measures based on shortest-paths routing (weighted and binary shortest paths) fall squarely in the periphery.

In the previous analyses, we examined synergies between pairs of predictors. Specifically, we focused on how synergies vary across the brain, which regions are associated the greatest improvements, and what pairs of predictors drive these improvements? First, we compared the increase in explained variance ($\Delta R^2$) as a result of including the second predictor. Here, when we calculate $\Delta R^2$, we adjust the multilinear model's $R^2$ to penalize for the addition of a second predictor (see Methods for details of the correction). As expected, all changes in variance explained were positive (Fig. 5d) and were largely concentrated in somatosensory systems (spin test, 1000 repetitions, false discovery rate fixed at $q = 0.05$; $p_{adj} = 0.0081$; Fig. 5e), suggesting that the biggest increases were associated with regions and systems whose baseline $R^2$ was among the greatest prior to introducing a second predictor.

For completeness, we all considered the effect of combining *all* predictors into the same linear model. Note that many of the

predictors are correlated, making it difficult to assess their unique contributions. However, we note that including all predictors yields $R^2$ values that are, on average, 3.3 ± 1.3 times greater than the best pairs of predictors. The regions that benefit the most from the inclusion of all predictors fall within default mode and salience/ventral attention systems, and, in some cases, exhibit increases in $R^2$ that are ten times that of the best predictor pair. The results of this additional analysis are summarized in Supplementary Fig. S6.

In parallel, we repeated the above analysis using principal components derived from predictors rather than the predictors themselves. This procedure helps address concerns related to correlated predictors. In general, the results of this analysis converge with those reported here. See Supplementary Fig. S7 for a summary of these results.

Collectively, these results demonstrate that improvements gained by using multiple predictors to explain FC have distinct spatial topography, favoring unimodal sensory systems. Moreover, even with multiple predictors, the predictability of FC in heteromodal cortices improves little.

**Structure-function coupling weakens across the human lifespan.** In the previous sections we systematically evaluated the utility of different structural predictors for explaining variance in regional patterns of FC. Those analyses were carried out using data from the Human Connectome Project and included subjects of, roughly, the same age range (young adult; 18–30 years). In this section, we use data from the enhanced Nathan Kline Institute-Rockland sample, which comprises 542 individuals from the Rockland, NY community whose ages range from childhood through senescence (7–85 years). Specifically, we focus on the magnitude of structure-function coupling across the lifespan and differences in the optimal predictor as a function of age.

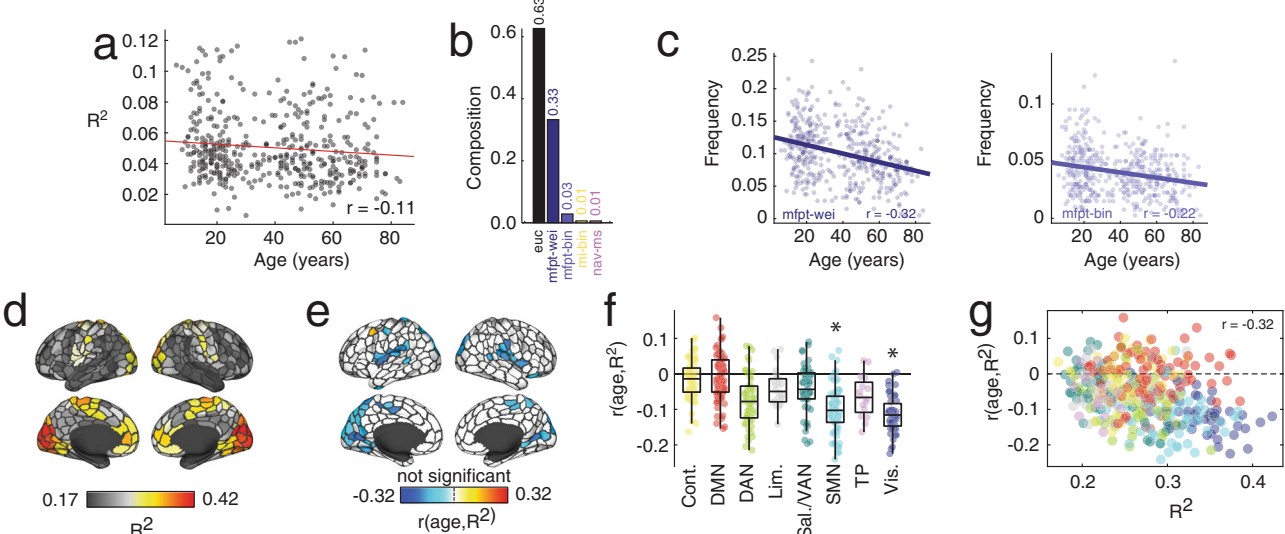

**Fig. 6 Lifespan variation in structure-function coupling based on communication models. a** Decreases in whole-brain variance explained across the lifespan. **b** Distribution of optimal predictors for each subject across the lifespan. **c** Correlations between age and the frequency with this weighted and binary mean first passage time were the optimal predictor of regions' FC patterns. **d** Whole-brain pattern of variance explained. **e** Correlation of variance explained with age projected onto the cortical surface. **f** Correlation of variance explained grouped by brain system (asterisks indicate statistical significance). Each point represents a brain region ($N = 400$ parcels defined based on[40]). In each boxplot, the "box" denotes interquartile range (IQR), the horizontal bar indicates the median value, and the whiskers include points that are within $1.5 \times IQR$ of upper and lower bounds of the IQR (25th and 7th percentiles). Any points that fall beyond the whiskers are, by convention, considered outliers. **g** Scatterplot of variance explained *versus* the correlation of variance explained and age. Source data are provided as a Source Data file.

First, we assessed how differences in global structure-function correspondence varied with age. To do this, we calculated the maximum $R^2$ for each participant across all predictors. Then, to rule out the possibility that inter-individual differences in variance explained is related to differences in sex, time of visit (for data acquisition), data quality measures like number of uncensored frames and the framewise displacement associated with those frames, or global network properties like total weight and binary density, we regressed these values out of each subjects' $R^2$ value. The residuals obtained following this procedure are, by definition, orthogonal to those nuisance variables. Finally, we calculated the linear correlation of these residuals with subjects' ages and observed that the two were significantly associated with one another ($r = -0.11$; $p = 0.02$); Fig. 6a, suggesting that the magnitude of structure-function correspondence decreases monotonically with age. Globally, the most common optimal predictors of FC were Euclidean distance (63% of participants) and weighted mean first passage time (33%) (Fig. 6b).

The previous analysis focused on global coupling between structure and function. Next, we investigated age-related differences in structure-function coupling at a local (regional) level. As with the global analysis, we regressed out the effect of sex, time of visit, the number of frames used to estimate FC, and the mean framewise displacement of "clean" frames along with global network properties. Because we were examining effects at the level of individual nodes, we also regressed out the effect of nodes' binary and weighted degrees. First, we asked whether the prevalence of certain predictors varied with age. For each region, we identified the predictor that best explained its regional pattern of FC and, for each subject, calculated the fraction of regions best explained by each factor. Then, we calculated the correlation of frequencies with biological age (Fig. 6c). We found that only weighted and binary mean first passage time were significantly correlated with age ($r = -0.32$ and $r = -0.22$; $p = 9.8 \times 10^{-13}$ and $p = 7.7 \times 10^{-7}$; false discovery rate fixed at $q = 0.05$; adjusted critical value of $p_{adj} = 0.0011$). We also repeated this analysis after

binning subjects according to their ages and found similar results irrespective of bin size (see Supplementary Fig. S8).

Next, we asked whether the maximum variance explained by any predictor – a measure of structure-function coupling – varied with age. We found a hemispherically symmetric (Fig. 6e) correlation pattern in which most regions, mirroring the global pattern, decreased with age. Interestingly, the spatial pattern of correlations was system-specific, with negative correlations significantly concentrated within somatomotor and visual systems (spin test, 1000 repetitions, false discovery rate fixed at $q = 0.05$; $p_{adj} = 0.00181$; Fig. 6f). Interestingly, the relationship between variance explained and its correlation with age was negative, so that regions with high levels of structure-function coupling in normative adults were more likely to decrease with age (Fig. 6g). Note that we also found system-specific correlations of $R^2$ with measures of intelligence, although these values were overall much weaker in magnitude and that, after multiple comparison corrections, none of the regional correlations are significant, although some system-level effects exist (spin test, 1000 permutations, $p_{adj} = 0.0054$; see Supplementary Fig. S9).

An important concern related to age differences in structure-function coupling is the possibility of a floor effect. Namely, that the $R^2$ of some regions cannot exhibit significant decreases because it is already near its theoretical floor value. To assess the likelihood of such an effect occurring, we estimated floor values for each region and subject using a permutation-based null model (100 repetitions; see Supplementary Fig. S10 for methodological details) and compared them against the observed $R^2$ values. In general, we found that most regions ($96.8 \pm 1.5\%$ were significantly greater than their theoretical floor) and that those regions nearing the floor did not overlap with the regions in which we reported strong age effects.

Collectively, these results suggest that the interrelationship of structural and functional connectivity covaries weakens with age. Notably, the areas that exhibit the greatest reductions fall within sensorimotor systems, which are among those with the strongest

coupling to begin with, and include regions in striate and extrastriate cortex, as well as primary motor cortex and superior parietal lobule. Areas within heteromodal systems, like default mode and control networks, on the other hand, exhibit subtle reductions in coupling magnitude and, in some cases, even increase with age (for instance, both dorsal and ventral prefrontal cortices along with temporal pole). Our findings point to heterogeneous differences in the complex relationship between the brain's physical wiring and its intrinsic functional organization.

## Discussion

Here, we aimed to address two questions about structure-function coupling in brain networks. What structurally-derived measure does the best job predicting FC? Second, how does the best measure vary locally, from one region to another? As a final application, we considered how the optimal predictor varies with biological age in a large lifespan dataset. We found that predictions of whole-brain FC were poor, irrespective of predictor, and could be improved upon by making predictions at a local, i.e. regional level. At this scale, the optimal predictor varied by region, with Euclidean distance and weighted mean first passage time among the best. Consistent with previous reports, the spatial patterning of structure-function coupling favored sensorimotor cortices. We then explored synergies between predictors and their inclusion in multi-linear models. Again, we found that Euclidean distance and weighted mean first passage time, along with measures of communicability and, in some instances, search information and path transitivity, exhibited strong synergistic relationships such that their joint inclusion in a model tended to support improvements in functional variance explained. Finally, we examined structure-function coupling across the human lifespan. We found that, globally, the magnitude of coupling decreased with age, an observation driven by decreases in local coupling of visual and somatomotor systems. Structure-function coupling in higher-order cognitive systems, including the control and default mode networks, went largely unchanged with age and, in some cases, even increased.

**Communication processes and sparse-to-fully weighted transformations of SC.** Many studies have attempted to link patterns of structural and functional connectivity to one another. At one extreme are studies that investigated neurobiologically realistic models, e.g. neural mass models (NMMs), whose parameters correspond to biophysical constants and generate time-varying voltage traces from neuronal populations[7,17,24,50,51]. These models offer a mechanistic description of how structural links constrain brain dynamics and give rise to cohesive and correlated activity. The performance of neural mass models can be improved upon by allowing for regional heterogeneity in parameters, matching additional features of empirical brain activity, including separation of dynamic timescales and the patterning of time-varying connectivity[52,53]. However, NMMs come at a high computational cost, limiting the possibility of performing exhaustive parameter searches or fitting the complete set of parameters at a subject-level.

At the other extreme are studies that link structure to function by directly comparing the weights of connections. In one of the earliest studies of the human connectome, the authors showed that structural weights (estimated as a length and volume normalized streamline count) and their corresponding functional connections are correlated, both globally and for select seed regions[2]. Similarly,[54] demonstrated that coherent patterns of spontaneous and task-evoked activity in the macaque oculomotor system are supported by anatomical connections. These types of correlative relationships are found at other spatial scales using

invasive reconstruction and imaging techniques applied to mode organisms. For instance, in[55], the authors used "barcoding" to reconstruct cellular-level synaptic connectivity and linked connection weights to the correlation structure of spontaneous activity recorded using widefield fluorescence imaging. Broadly, these approaches demonstrate that, for structurally connected neural elements, their anatomical connection weight is correlated with the similarity of their recorded activity. This approach for studying structure-function relationships, however, is limited in that comparisons between connection weights can only be carried out for directly connected pairs of neural elements. If two cells, populations, or regions are not directly connected, then it becomes impossible to make a prediction about its functional coupling.

The results reported here are situated between these two extremes and adopt useful principals from each while avoiding some of the pitfalls[12]. Specifically, we focus on predicting the weights of functional connections by transforming the sparse SC matrix into a fully-weighted (and possibly signed) matrix. Some of these transformations incorporate elements of dynamics. Flow graphs, for example, embed the probabilistic flow of random walkers between two nodes into the edges of a graph[41]. Other transformations embody communication policies that the brain could conceivably implement as means of transmitting a signal/information from one region to another. Shortest paths are an example of a centralized communication process, in that to take advantage of these paths would require global knowledge (a centralized pool of information) of the network's shortest path structure. In contrast, decentralized processes like diffusion/random walks or navigation evolve without the need for any additional information[11,45]. Path transitivity and search information blend these two concepts, treating shortest paths as fixed constructs, but asking how easily they could be traversed passively by a knowledgeless random walker. Other transformations represent geometric relationships between nodes or the topological similarity of their structural connectivity patterns.

Practically, using matrix-based predictors to explain FC has many advantages. Unlike biophysical models, all of the predictors studied here can be generated in seconds of computation time, reducing the computational burden associated with realistic models. However, this improvement in computational complexity does not require that we jettison all information about dynamics; as noted earlier, many of the matrices provide summary information about dynamical processes. This approach also circumvents the issue of missing connections. Unlike correlative methods that can only compute the similarity of existing structural and functional connections, this approach transforms the sparse structural matrix into a fully-weighted matrix, where every entry can be, in principle, non-zero. This allows for a more direct comparison. Lastly, previous studies of identical datasets have shown that the matrix predictors tend to outperform even biophysical models in terms of matching empirical patterns of FC[8,11].

Here, we extend the matrix-based prediction approach from a global, whole-brain level to a the level of individual brain regions. Our approach builds upon other recent studies that examined heterogeneity in coupling patterns across regions[30,31] and systems[48]. Despite differences in dataset, parcellation, and choice of predictor, our results are closely aligned with those of previous studies, which reported strong coupling in sensorimotor cortices and weaker coupling in heteromodal regions. Interestingly, heteromodal cortex includes regions that expanded dramatically over the course of mammalian evolution[56] and others that develop at a protracted rate[57], leading to the hypothesis that their interregional connectivity may be shaped by different organizational principles than regions in unimodal cortex and serving as a

possible explanation for the weakened structure-function coupling.

Of course, there are serious tradeoffs associated with modeling FC in terms of matrix-based predictors derived from SC. Namely, it sacrifices the neurobiological plausibility of NMMs for computational ease and the interpretability of direct structure/function correlations in order to generate fully-weighted matrices. Additionally, matrix-based predictors fundamentally lack a temporal dimension. That is, they compress information about dynamical processes, e.g. diffusion, navigation, shortest paths routing, into matrix form, the precise temporal evolution of those processes is lost. In contrast, biophysical models document the temporal evolution of activity, generating spike trains, voltage traces, or hemodynamic signals from cells, populations, or regions[58–60]. These temporal data provide an additional target for modeling studies; rather than simply matching the correlation structure of brain activity, can a model also replicate its time-varying features? Indeed, while recent work has begun to investigate time-varying structure-function coupling[53,61–63], future studies are necessary.

In summary, there exists a spectrum of methods for assessing and modeling structure-function coupling in empirical data[18,64–70]. Each method possesses a distinct set of advantages and disadvantages, such that some research questions are better addressed by one method and not another. For instance, biophysical models offer a neurobiological account of the relationships between structure and function while making population-level time courses available for analysis. On the other hand, the complexity of biophysical models might make them unappealing for studies where the aim is to generate a biomarker of structure-function coupling. In that case, a simple correlation between SC and FC weights may suffice. In general, these methods should be viewed as complementary rather than adversarial, affording researchers the opportunity to examine data under different sets of assumptions, across spatial scales, with varying levels of neurobiological realism, and using different computational tools.

**Shortest paths or diffusion; centralized or decentralized processes?** Path-based metrics are often used as markers to compare populations of individuals in an effort to distinguish clinical subjects from controls[71] or to be linked with a continuous measure, e.g. intelligence scores[72].These metrics include the well-known characteristic path length – the mean number of steps in shortest paths over all pairs of nodes – and efficiency, the mean reciprocal of the shortest path lengths[73].

Although these measures are commonplace in network neuroscience, they implicitly prioritize shortest paths as the communication routes between brain regions. Although superficially this seems like a reasonable assumption, other studies have cast doubt on these findings, noting that the backbone of shortest paths involves only small fraction of network edges[21,74] and that, for a brain to use shortest paths for communication, requires that it has global knowledge of its shortest path structure[11,22], which is not obviously biologically plausible.

Recently, however, a growing number of studies have presented alternative and decentralized communication models. These include models of diffusion and random walk dynamics[24,75], epidemic spread[21,76], more complicated models that allow for an interpolation between centralized and decentralized processes[22], and navigation models[42,77]. Unlike shortest paths, these communication processes evolve using local knowledge – the next step depends on the weights of edges (random walks) or the distance of directly connected neighbor from the eventual target.

Here, and in agreement with other studies[11], we find that shortest paths structure (both binary and weighted) do a poor job

in recapitulating patterns of FC. In contrast, measures like mean first passage time, communicability (which emphasizes not just the single shortest path in a network, but *all* walks of *all* lengths), and others derived from navigation appear near the top of the list in terms of frequency. Although these observations do not conclusively demonstrate that shortest path structure plays no role in communication processes, they do support the hypothesis that decentralized measures may play an outsized (and possibly underappreciated) role in shaping interregional communication processes. We note, however, that under certain circumstances, e.g. biased random walks or navigation strategies[42], may access a network's shortest paths.

**The impact of space on structure-function coupling.** One of the long-standing observations about brain networks is that their architecture is shaped, in part, by the space in which they are embedded[78]. The tight statistical relationship between distance and presence/absence of structural connections and their weights holds at virtually all scales[74], from cellular-level connectomes[79], to areal maps[80], to non-invasive imaging with MRI[81]. This relationship also holds for functional imaging data[82], although the coupling between space and FC is statistically less severe[83,84].

The observation that SC and FC are both constrained by space has lead to speculation that at least some fraction of variance in structure-function coupling can be attributed to their joint embedding in Euclidean space. Indeed, previous studies have shown that structure-function correlations are attenuated after regressing out the effects of Euclidean distance[11,23], though the resulting correlations suggest that there remains some residual relationship.

Here, we adopted a local perspective on structure-function relationships and directly compared different predictors with one another. This allows us to test the extent to which Euclidean distance outperformed any of the other network measures for predicting FC. Indeed, we found that Euclidean distance was always among the best measures. This was true for predicting FC globally in both the HCP and NKI datasets, as well as the analysis of local connectivity. However, our local analysis revealed that, for many regions, especially those in control, somatomotor, and temporoparietal networks, Euclidean distance was outperformed by other measures. On the other hand, Euclidean distance was overrepresented as an optimal predictor within the visual, salience/ventral attention, and dorsal attention networks.

These observations suggest that the impact of spatial relationships on SC an FC and their statistical coupling to one another is heterogeneous and, to some extent, system-specific.

**Differences in structure-function coupling with age.** The progression through development, maturation, and senescence is one of the most profound and shared human experiences. It is accompanied by increased diffusivity of task-evoked brain activity[85] and reductions in system segregation[35,36]. Similar differences occur structurally, with reductions in modularity[86] and increased characteristic path length[32]. Less is known about how structural and function differences occur in parallel, and especially at the local scale[30].

Here, we investigate lifespan differences in structure-function coupling, and find that with age, the global correlation is reduced. At the local scale, these differences are paralleled by reductions in both weighted and binary measures of mean first passage time, whichrefers to the number of steps in a random walk when a random walker starting at node $i$ is likely to have visited node $j$. Importantly, mean first passage time is a descriptor for a diffusive (random walk) process. Based on these findings and along with the observation that the magnitude of global structure-function

coupling decreases with age, we speculate that, decentralized patterns of interregional communication may degrade over the human lifespan, prompting a decoupling of functional connectivity from structure.

However, there are several potential limitations of this analysis. Specifically, there exist alternative interpretations of our results that are difficult to rule out conclusively. Here, we interpret $R^2$ as a measure of how strongly or weakly structural and functional connectivity are coupled to one another and that age-related differences in $R^2$ reflect shifts in network-level communication patterns or policies. Aging, however, is accompanied by systematic differences in other brain and physiological measurements[87], including level of in-scanner motion[35,88], respiratory and cardiac signatures, and neurovascular coupling[89]. Here, we adopted a conservative approach to addressing motion-related issues, excluding from analysis any subjects whose in-scanner motion exceeded a threshold. This approach, however, may inadvertently exclude subjects with usable data. Addressing issues related the neurovascular coupling, on the other hand, presents a greater challenge, as how best to assess and correct for subject- and region-level coupling patterns remains an open frontier in aging neuroscience[90–92]. This topic should be the focus of future studies.

**Future directions and limitations**. This study suffers from a number of limitations. Streamline tractography provides an estimation of the underlying white matter that is potentially hampered by biases involving complex fiber geometry and limitations given the inverse problem that the methodology aims to solve[93]. For instance, our current study focuses on cortico-cortical pathways, excluding subcortical nuclei and the cerebellum from analysis. While the justification for this exclusion is straightforward – limitations of whole-brain diffusion imaging and tractography make it challenging to accurately resolve the connections of small structures in the basal ganglia[94,95] – it nonetheless leaves open the possibility that our results will change with the inclusion of these structures. Future studies should examine whole-brain communication processes and include structures from the subcortex and cerebellum[96].

Another potential limitation concerns the breadth of matrix-wide predictors. Here, we examine 40 predictors, each of which of represent a dynamical, topological, or geometric factor that could explain patterns of FC. However, there are other predictors that could, in principle, be explored here but due to the scope of the paper are not. In addition, there are other methods, including spectral decompositions[69], deep learning[67], and embedding models[70,97], that can lead to high levels of structure-function coupling, but present little mechanistic insight. Future studies should investigate alternative predictors and other methods in greater detail, seeking to merge machine-learning and neurobiological accounts of structure-function coupling.

Interestingly, while the regional patterns of structure-function coupling in NKI and HCP datasets were correlated, the coupling patterns were not identical. While the origins of this difference remain unclear, possible explanations include differences in sample size ($N = 95$ in HCP *versus* $N = 542$ in NKI, before further exclusion for motion), heterogeneity, and community representativeness. Other possibilities include differences in data quality and preprocessing strategies. In the NKI dataset, functional and structural networks were estimated using a volumetric parcellation in subject anatomical space. In contrast, the HCP data was processed using surface-based analyses. Indeed, recent studies have identified notable differences between these two image processing pipelines[98]. Future studies should investigate this possibility in greater detail, documenting the effect of different processing pipelines on local structure-function coupling.

A final limitation concerns the possibility that the reported correlations between $R^2$ and age could be attributed to a floor effect. That is, any regions that start with low $R^2$ early in life necessarily have less room to decrease over the course of the lifespan, whereas regions that start with high levels of $R^2$ have more room to decrease. Although we show that the fraction of regions that may be susceptible to this effect is small, the spatial pattern is correlated with the reported $R^2$ *vs* age correlation map (see Supplementary Fig. S10). Disentangling true age-related differences from floor effects is challenging and may require the use of an unbounded measure other than $R^2$ for assessing structure-function correspondence. Future work should investigate these potential confounds in greater detail.

## Methods

### Datasets

*Human connectome project*. The Human Connectome Project (HCP) dataset[38] consisted of structural magnetic resonance imaging (T1w), resting state functional magnetic resonance imaging (fMRI) data, as well as diffusion magnetic resonance imaging data (dMRI) from 100 unrelated adult subjects. These subjects were selected as they comprised the "100 Unrelated Subjects" released by the Human Connectome Project. After excluding subjects based on data completeness and quality control (see Quality Control), the final subset utilized included 95 subjects (56% female, mean age = 29.29 ± 3.66, age range = 22–36). The study was approved by the Washington University Institutional Review Board and informed consent was obtained from all subjects. Participants were not compensated. A comprehensive description of the imaging parameters and image preprocessing can be found in[99]. Images were collected on a 3T Siemens Connectome Skyra with a 32-channel head coil. Subjects underwent two T1-weighted structural scans, which were averaged for each subject (TR = 2400 ms, TE = 2.14 ms, flip angle = 8°, 0.7 mm isotropic voxel resolution). Subjects underwent four resting state fMRI scans over a two-day span. The fMRI data was acquired with a gradient-echo planar imaging sequence (TR = 720 ms, TE = 33.1 ms, flip angle = 52°, 2 mm isotropic voxel resolution, multiband factor = 8). Each resting state run duration was 14:33 min, with eyes open and instructions to fixate on a cross. Finally, subjects underwent two diffusion MRI scans, which were acquired with a spin-echo planar imaging sequence (TR = 5520 ms, TE = 89.5 ms, flip angle = 78°, 1.25 mm isotropic voxel resolution, b-vales = 1000, 2000, 3000 s/mm², 90 diffusion weighed volumes for each shell, 18 b = 0 volumes). These two scans were taken with opposite phase encoding directions and averaged.

*Nathan Kline Institute, Rockland Sample*. The Nathan Kline Institute Rockland Sample (NKI) dataset consisted of structural magnetic resonance imaging, resting state functional magnetic resonance imaging data, as well as diffusion magnetic resonance imaging data from 811 subjects (downloaded December 2016 from the INDI S3 Bucket) of a community sample of participants across the lifespan. After excluding subjects based on data and metadata completeness and quality control (see Quality Control), the final subset utilized included 542 subjects (56% female, age range = 7–84). The study was approved by the Nathan Kline Institute Institutional Review Board and Montclair State University Institutional Review Board and informed consent was obtained from all subjects. Subjects were compensated for participation. A comprehensive description of the imaging parameters can be found online at the NKI website. Briefly, images were collected on a Siemens Magneton Trio with a 12-channel head coil. Subjects underwent one T1-weighted structural scan (TR = 1900 ms, TE = 2.52 ms, flip angle = 9°, 1 mm isotropic voxel resolution). Subjects underwent three differently parameterized resting state scans, but only one acquisition is used in the present study. The fMRI data was acquired with a gradient-echo planar imaging sequence (TR = 645 ms, TE = 30 ms, flip angle = 60°, 3 mm isotropic voxel resolution, multiband factor = 4). This resting state run lasted approximately 9:41 seconds, with eyes open and instructions to fixate on a cross. Subjects underwent one diffusion MRI scan (TR = 2400 ms, TE = 85 ms, flip angle = 90°, 2 mm isotropic voxel resolution, 128 diffusion weighted volumes, b-value = 1500 s/mm², 9 b = 0 volumes).

**Quality control**. For HCP, all preprocessed time series were visually inspected for visual artifact. Subject motion measurements during the fMRI and DWI scanning sessions were obtained from the HCP minimal preprocessing pipeline output directories (files: `Movement_RelativeRMS.txt` and `eddy_unwarped_i-mages.eddy_movement_rms`). Across fMRI sessions and the single fMRI session, the mean and mean absolute deviation of the motion measurements were calculated, resulting in four summary motion measures per subject. Subjects exceeding 1.5 times the inter-quartile range (in the adverse direction) of the measurement distribution for more than one of these summary motion measurements were excluded. This resulted in the exclusion of four subjects. One additional subject was excluded due to software error during DWI processing.

The NKI was downloaded in December of 2016 from the INDI S3 Bucket. At the time of download, the dataset consisted of 957 T1w (811 subjects), 914 DWI

(771 subjects), and 718 fMRI ("acquisition645"; 634 subjects) images. T1w and DWI images, and tractography results were first filtered based on visual inspection. T1w images were filtered based on artifact, such as ringing or ghosting (43 images) and for FreeSurfer reconstruction failure (105 images) as assesses with the ENIGMA QC tools, leaving 809 T1w images (699 subjects). DWI images were filtered based on corrupt data (13 images) and artifact on fitted fractional anisotropy maps (18 images), leaving 883 images (747 subjects). Tractography was run on 781 images (677 subjects) that had both quality controlled T1w and DWI images. Tractography results were filtered based on artifact, which include failure to resolve callosal, cingulum, and/or corticospinal streamlines or errors resulting in visually sparse streamline densities, resulting in 764 tractography runs (661 subjects). T1w, DWI, and fMRI images were then filtered using computed image quality metrics[100–102]. T1w images were excluded if the scan was marked as an outlier (1.5x the inter-quartile range in the adverse direction) in three or more of following quality metric distributions: coefficient of joint variation, contrast-to-noise ratio, signal-to-noise ratio, Dietrich's SNR, FBER, and EFC. DWI images were excluded if the percent of signal outliers, determined by eddy_qc, was greater than 15%. Furthermore, DWI were excluded if the scan was marked as an outlier (1.5x the inter-quartile range in the adverse direction) in two or more of following quality metric distributions: temporal signal-to-noise ratio, mean voxel intensity outlier count, or max voxel intensity outlier count. fMRI images were excluded if greater than 15% of time frames exceeded 0.5 mm framewise displacement. Furthermore, fMRI images were excluded the scan was marked as an outlier (1.5x the inter-quartile range in the adverse direction) in 3 or more of the following quality metric distributions: DVARS standard deviation, DVARS voxel-wise standard deviation, temporal signal-to-noise ratio, framewise displacement mean, AFNI's outlier ratio, and AFNI's quality index. This image quality metric filtering excluded zero T1w images, 16 DWI images, and 21 fMRI images. Following this visual and image quality metric filtering, 809 T1w images (699 subjects), 728 DWI images (619 subjects), and 697 fMRI images (633 subjects). The intersection of subjects with at least one valid T1w, DWI, and fMRI images totaled 567 subjects. Finally, age metadata was available for 542 of these subjects.

**Image processing**. Structural, functional, and diffusion images of the HCP dataset were minimally preprocessed according to the description provided in[99]. Briefly, T1w images were aligned to MNI space before undergoing FreeSurfer's (version 5.3) cortical reconstruction workflow. fMRI images were corrected for gradient distortion, susceptibility distortion, and motion, and then aligned to the corresponding T1w with one spline interpolation step. This volume was further corrected for intensity bias and normalized to a mean of 10000. This volume was then projected to the 32k_fs_LR mesh, excluding outliers, and aligned to a common space using a multi-modal surface registration[103]. The resultant CIFTI file for each HCP subject used in this study followed the file naming pattern: *_REST{1,2}_{LR,RL} _Atlas_MSMAll.dtseries.nii. DWI images were normalized to the mean b0 image, corrected for EPI, eddy current, and gradient non-linearity distortions, and motion, and aligned to subject anatomical space using a boundary-based registration[104]. In addition to HCP's minimal preprocessing, diffusion images were corrected for intensity non-uniformity with N4BiasFieldCorrection[105]. FSL's dtifit was used to obtain scalar maps of fractional anisotropy, mean diffusivity, and mean kurtosis. The Dipy toolbox (version 1.1)[106] was used to fit a multi-shell multi-tissue constrained spherical deconvolution[107] to the diffusion data with a spherical harmonics order of 8, using tissue maps estimated with FSL's fast[108]. Tractography was performed using Dipy's Local Tracking module[106]. Multiple instances of probabilistic tractography were run per subject[109], varying the step size and maximum turning angle of the algorithm. Tractography was run at step sizes of 0.25 mm, 0.4 mm, 0.5 mm, 0.6 mm, and 0.75 mm with the maximum turning angle set to 20°. Additionally, tractography was run at maximum turning angles of 10°, 16°, 24°, and 30° with the step size set to 0.5 mm. For each instance of tractography, streamlines were randomly seeded three times within each voxel of a white matter mask, retained if longer than 10 mm and with valid endpoints, following Dipy's implementation of anatomically constrained tractography[110], and errant streamlines were filtered based on the cluster confidence index[111].

For NKI, T1w images were submitted to FreeSurfer's cortical reconstruction workflow (version 6.0). The FreeSurfer results were used to skull strip the T1w, which was subsequently aligned to MNI space with 6 degrees of freedom. fMRI preprocessing was performed using the fMRIPrep version 1.1.8[13]. The following description of fMRI preprocessing is based on fMRIPrep's documentation. This workflow utilizes ANTs (2.1.0), FSL (5.0.9), AFNI (16.2.07), FreeSurfer (6.0.1), nipype[112], and nilearn[113]. Each T1w was corrected using N4BiasFieldCorrection[105] and skull-stripped using antsBrainExtraction.sh (using the OASIS template). The ANTs derived brain mask was refined with a custom variation of the method to reconcile ANTs-derived and FreeSurfer-derived segmentations of the cortical gray-matter of Mindboggle[114]. Brain tissue segmentation of cerebrospinal fluid (CSF), white-matter (WM) and gray-matter (GM) was performed on the brain-extracted T1w using fast[108]. Functional data was slice time corrected using 3dTshift from AFNI and motion corrected using FSL's mcflirt. "Fieldmap-less" distortion correction was performed by co-registering the functional image to the same-subject T1w with intensity inverted[115] constrained with an average fieldmap template[116], implemented with antsRegistration. This was followed by co-registration to the

corresponding T1w using boundary-based registration[104] with 9 degrees of freedom, using bbregister. Motion correcting transformations, field distortion correcting warp, and BOLD-to-T1w transformation warp were concatenated and applied in a single step using antsApplyTransforms using Lanczos interpolation. Frame-wise displacement[117] was calculated for each functional run using the implementation of Nipype. The first four frames of the BOLD data in the T1w space were discarded. Diffusion images were preprocessed following the "DESIGNER" pipeline using MRTrix (3.0)[118,119], which includes denoising, Gibbs ringing and Rician bias correction, distortion and eddy current correction[120] and B1 field correction. DWI were then aligned to their corresponding T1w and the MNI space in one interpolation step with B-vectors rotated accordingly. Local models of white matter orientation were estimated in a recursive manner[121] using constrained spherical deconvolution[107] with a spherical harmonics order of 8. Tractography was performed using Dipy's Local Tracking module[106]. Probabilistic streamline tractography was seeded five times in each white matter voxel. Streamlines were propagated with a 0.5 mm step size and a maximum turning angle set to 20°. Streamlines were retained if longer than 10 mm and with valid endpoints, following Dipy's implementation of anatomically constrained tractography[110].

**Network definition**
*Parcellation*. As HCP fMRI was provided in 32k_fs_LR space, this data could be parcellated based on the available Schaefer 400 parcellation[40] in the CIFTI file format. For HCP DWI and NKI fMRI and DWI, the Schaefer 400 parcellation was rendered as a volumetric parcellation in each subject's anatomical space within the gray matter ribbon. To transfer the parcellation from fsaverage to subject space, FreeSurfer's mris_ca_label function was used in conjunction with a pre-trained Gaussian classifier surface atlas[122] to register cortical surfaces based on individual curvature and sulcal patterns.

*Structural connectivity*. For HCP, for each tractography instance, streamline counts were normalized by dividing the count between nodes by the geometric average volume of the nodes. Since tractography was run nine times per subject, edge values were collapsed across runs. To do this, the weighted mean was taken with weights based on the proportion of total streamlines at that edge. This amounts to calculating the expected value, where probabilities are based on the proportion of total edge weight across tracotgraphy instances. This operation biases edge weights towards larger values, which reflect tractography instances better parameterized to estimate the geometry of each connection. For NKI, streamline counts were normalized by dividing the count between nodes by the geometric average volume of the nodes.

*Functional connectivity*. For HCP and NKI, each preprocessed BOLD image was linearly detrended, band-pass filtered (0.008–0.08 Hz), confound regressed and standardized using Nilearn's signal.clean function, which removes confounds orthogonally to the temporal filters. The confound regression strategy included six motion estimates, mean signal from a white matter, cerebrospinal fluid, and whole brain mask, derivatives of these previous nine regressors, and squares of these 18 terms. Spike regressors were not applied to the HCP data. Spike regressors for frames with motion greater than 0.5 mm framewise displacement were applied to the NKI data. The 36 parameter strategy (with and without spike regression) has been show to be a relatively effective option to reduce motion-related artifacts[123]. Following these preprocessing operations, the mean signal was taken at each node, in either the surface space (HCP) or volumetric anatomical space (NKI).

*Frame censoring and exclusion criteria*. In the main text, we analyzed data that had been processed using the above procedure. We also performed extensive post-processing of these data to reduce the likelihood that in-scanner motion contributed to any reported effects[117,123–125]. Specifically, we implemented the following steps for both the HCP and NKI datasets:

1. Using the Movement_RelativeRMS.txt time series, we identified frames with motion greater than 0.15. These frames were immediately censored and not used in the estimation of FC.
2. We also censored any low-motion time points that were within two frames of any of the frames censored in step 1.
3. Following steps 1 and 2, we further censored any sequence of temporally contiguous low-motion frames that was shorter than five frames.
4. Lastly, we excluded a scan if more than 50% of its frames were flagged as high-motion following steps 1–3, i.e. were censored. If any scan from a given subject was removed, we removed that subject and all of their other scans from analysis.

Given these criteria, we retained 70/95 HCP subjects. Of these remaining subjects, we retained, on average, 90.3 ± 0.97% of their frames. The frames that were retained had 0.07 ± 0.01 relative motion. In contrast, the censored frames had relative motion of 0.15 ± 0.03. For the NKI dataset, we retained 474/542 subjects, keeping 93.3 ± 10.0% of frames, with a mean motion level of 0.06 ± 0.01. The censored frames had a motion level of 0.16 ± 0.06. In the Supplementary Material we explored the relationship of structure-function coupling with age after binning subjects based on their biological age. Using 10 bins of approximately equal numbers of subjects, we found that the subjects excluded due to frame censoring

disproportionately impacted older age bins (correlation between the number of subjects excluded in each bin and the bin's median age; $\rho = 0.47$; $p = 0.038$). All results reported in the main text were generated using these datasets.

## Predictors

*Flow graphs.* A flow graph is a transformation of a network's (possibly sparse) connectivity matrix, $A_{ij}$, into a fully-weighted matrix in which the dynamics of a Markov process are embedded into edge weights[41]. Flow graphs have been applied in neuroscience for the purposes of community detection[126]. For a continuous time random walk with dynamics $\dot{p}_i = -\sum_j L_{ij} p_j$, the corresponding flow graph is given by $A'(t)_{ij} = (e^{-tL})_{ij} s_j$. In these expressions, the matrix $L$ is the normalized Laplacian whose elements are given by $L_{ij} = D - A/s$, where $s_i = \sum_j A_{ij}$ is a node's degree or weighted degree and $D$ is the degree diagonal matrix (a square matrix the elements of $s$ along its diagonal). The variable $p_i$ represents the probability of finding a random walker on vertex $i$.

The element $A'(t)_{ij}$ represents the probabilistic flow of random walkers between nodes $i$ and $j$ at time $t$. Here, we generated flow graphs using both binary and weighted structural connectivity matrices at evaluated them at different Markov times, $t$. Specifically, we focused on $t = 1, 2.5, 5,$ and $10$. We refer to these variables as *fgbin-* or *fgwei-* followed by *Markov time, t*.

*Navigation.* The aim of many networks is to move something from one point in the network to another in as few steps as possible, i.e. to take advantage of shortest paths. However, doing so requires requires full knowledge of a network's shortest path structure, which may not be a realistic assumption, especially for naturally-occurring biological systems like brains. However, it may be the case that simple routing strategies – rules or heuristics for how to move from one node to another – can sometimes uncover optimal or near-optimal shortest paths. One such routing rule is, given a target node $T$, to always move towards the node nearest the target in some metric space, e.g. Euclidean space.

Recently, this navigation approach was applied to brain networks[42]. This study defined two measures based on navigation of connectome data. First, they defined the number of hops in the shortest path uncovered by the navigation process. We refer to this variable as *nav-num*. Note that for some node pairs, the navigation procedure leads to a dead end or a cycle – in which case the number of hops is listed as $\infty$. For the completed paths, the authors also defined their total length in metric space (in this case Euclidean distance). We refer to this variable as *nav-ms* and, like *nav-num*, impute incomplete paths with values of $\infty$.

*Communicability.* Communicability[43] is a weighted sum of walks of all lengths between pairs of nodes. For a binary network, it is calculated as $G = e^A$ or $\sum_{p=0}^{\infty} \frac{A^p}{p!}$. The contribution of direct links (1-step walks) is $\frac{A^1}{1!}$, two-step walks is $\frac{A^2}{2!}$, three-step is $\frac{A^3}{3!}$, and so on. In other words, longer walks have larger denominators and, effectively, are penalized more severely. We denote this measures as *comm-bin*.

For weighted networks, we follow[44] and first normalize the weighted connectivity matrix as $A' = D^{-1/2} A D^{-1/2}$ where $D$ is the degree diagonal matrix. As before, this normalized matrix is the exponentiated to calculate the weighted communicability $G_{wei} = e^{A'}$. We denote this measures as *comm-wei*.

*Matching index.* The matching index[26] is a measure of overlap between pairs of nodes based on their connectivity profiles. Suppose $\Gamma_i = j: A_{ij} > 0$ is the set of all nodes directly connected to node $i$. We can calculate the matching index between nodes $i$ and $j$ as $M_{ij} = \frac{|\Gamma_{i\backslash j} \cap \Gamma_{j\backslash i}|}{|\Gamma_{i\backslash j} \cup \Gamma_{j\backslash i}|}$. Here, $\Gamma_{i\backslash j}$ refers to the neighbors of node $i$ excluding node $j$.

*Shortest paths.* In a network, each edge can be associated with a cost. For binary networks, the cost is identical for each edge; for weighted networks the cost can be obtained by a monotonic transformation of edges' weights to length, e.g. by raising an edge's weight to a negative power. The shortest path between a source node, $s$, and a target node, $t$, is the sequence of edges $\pi_{s \to t} = \{A_{si}, A_{ij}, ..., A_{kt}\}$ that minimizes the sum $C_{si} + C_{ij} + ... + C_{kt}$, where $C_{si}$ is the cost of traversing the edge linking nodes $s$ and $i$.

Here, we calculated shortest paths matrices for the binary network (where the cost is identical for all existing edges) and also for a parameterized affinity-to-cost transformation evaluated at several different parameter values. Specifically, we used the following transformation: $C_{ij} = A_{ij}^{-\gamma}$. We focused on the parameter values $\gamma = 0.125, 0.25, 0.5, 1.0, 2.0,$ and $4.0$. We refer to these measures as *pl-bin* and *pl-wei-* followed by $\gamma$ value.

*Cosine similarity.* The cosine similarity measures the angle between two vectors, $x = [x_1, ..., x_P]$, and $x = [y_1, ..., y_P]$. Specifically, it measures $S_{xy} = \frac{x \cdot y}{\|x\| \cdot \|y\|}$. Here, we treated regions' connectivity profiles (the row of the connectivity matrix) as vectors and computed the similarity between all pairs of regions. We repeated this procedure for both the binary (*cos-bin*) and weighted (*cos-wei*) connectivity matrices.

*Search information.* Search information measures the amount of information (in bits) required to traverse shortest paths in a network[11,45]. If the shortest path

between nodes $s$ and $t$ is given by $\pi_{s \to t} = \{s, i, j, ..., k, l, t\}$, then the probability of taking that path is given by: $P(\pi_{s \to t}) = p_{si} \times p_{ij} \times ... \times p_{kl} \times p_{lt}$, where $p_{ij} = \frac{A_{ij}}{\sum_j A_{ij}}$. The information required to take this path, then, is $S(\pi_{s \to t}) = \log_2[P(\pi_{s \to t})]$.

Here, we calculated search information based on binary shortest paths (*si-bin*) and based on shortest paths obtained from each of the weight-to-cost transformations (*si-wei-$\gamma$ value*).

*Mean first passage time.* The mean first passage time (MFPT) refers to the expected number of steps a random walk must evolve for a random walked starting at node $i$ to end up at node $j$[46,127]. Here, we expressed the columns as z-scores to remove nodal (column) biases and analyzed the resulting matrices for the binary (*mfpt-bin*) and weighted (*mfpt-wei*) connectivity matrices.

*Euclidean distance.* The final predictor that we considered was the Euclidean distance between regional centers of mass (*euc*).

## Core-periphery analysis

We used a core-periphery model to analyze the count matrix of how often pairs of predictors were included together in the same multilinear model. In this context, a *core* refers to a group of predictor that are densely internally connected and to a *periphery*, that connect to the core but not to other peripheral predictors[128]. To identify core-periphery structure, we used a variant of a common core-periphery definition in which the transition from core to periphery varies smoothly. Rather than using a binary assignment of nodes to a core or a periphery, this allows nodes to have a graded and continuous assignments. We begin by defining the $N \times 1$ vector $C_i$ of non-negative elements[129]. Given this vector, we then defined the matrix $C_{ij} = C_i C_j$ subject to the constraint that $\sum_{ij} C_{ij} = 1$. The values in the vector $C$ are permutations of the vector:

$$C_m^* = \frac{1}{1 + exp(-(m - \beta N) \times tan(\pi\alpha/2))}. \tag{1}$$

The coreness of each node is the permutation of $C_m^*$ that maximizes the core quality function:

$$R = \sum_{ij} G_{ij} C_i C_j. \tag{2}$$

This method introduces two free parameters, $\alpha \in [0, 1]$ and $\beta \in [0, 1]$. The value of $\alpha$ determines the sharpness of the core-periphery boundary. With $\alpha = 1$, the transition is binary while the transition with $\alpha = 0$ is maximally fuzzy. Similarly, the value of $\beta$ determines the size of the core; as $\beta$ ranges from 0 to 1, the size of the core varies from $N$ to 0. In our application, we performed a grid search of 51 logarithmically-spaced values of $\alpha$ and $\beta$, using a simulated annealing algorithm to maximize $R$ (with 25 restarts).

## Community detection

In the main text, we described an analysis in which we clustered brain regions based on the similarity of their optimal predictor. Briefly, this procedure entailed calculating for each brain region the frequency with which predictor, $p$, was optimal, i.e. explained the greatest amount of variance in that regions' FC pattern. This resulted in a vector $\mathbf{h} = \{h_1, ..., h_p, ..., h_{40}\}$ subject to the constraint that $\sum_p h_p = 1$. We then computed the correlation between all pairs of brain regions based on these vectors. We refer to this matrix as $S$, whose element $S_{ij}$ denotes the similarity between feature vectors of regions $i$ and $j$.

To better understand the structure of $S$, we clustered brain regions into communities using modularity maximization[5,130,131]. To do so, we optimized the modularity quality function:

$$Q = \sum_{ij} B_{ij} \delta(\sigma_i, \sigma_j) \tag{3}$$

where $B_{ij} = S_{ij} - \gamma \cdot P_{ij}$. In this expression, $P_{ij}$ is the expected weight of the connection between regions $i$ and $j$ and $\gamma$ is a structural resolution parameter that tunes the number of size of detected communities. For simplicity, we set $\gamma = \langle S \rangle = 0.296 \approx 0.3$ and used this value for all pairs of brain regions. For completeness, however, we also tested other resolution parameter values, ranging from $\gamma = 0$ to $\gamma = 1$ in increments of 0.025. We show some of these communities in Supplementary Fig. S4d.

We used a generalization of the Louvain algorithm[132] to optimize $Q$. This algorithm is non-deterministic and results in a degeneracy of near-optimal solutions. To resolve this degeneracy, we used a consensus clustering algorithm in which we ran the Louvain algorithm 1000 times (random initial conditions) and computed the co-assignment probability for all pairs of brain regions, i.e. the likelihood that they were assigned to the same community[74,133–135]. Then, we calculated the expected probability that any two nodes were assigned to the same community after randomly and independently permuting the order of each of the 1000 partitions. From these two values, we calculated a new modularity matrix – the observed co-assignment probability minus the expected – and clustered this matrix again (repeating the algorithm 1000 times). This sequence – modularity maximization followed by construction of observed and expected co-assignment probabilities – was repeated until each of the 1000 runs converged to an identical solution. At this point the consensus algorithm terminated.

**Adjusted $R^2$ for the multilinear model**. In the main text we reported the change in variance explained, $\Delta R^2$ as a result of including two predictors. In general, we might expect increases in $R^2$ simply due to the inclusion of a second parameter. One strategy to account for this addition is to adjust the $R^2$ measure based on the number of predictors. Specifically, we calculated $R^2_{adjusted} = 1 - \frac{(1-R^2)(N-1)}{N-p-1}$ for each region for both the one- and two-term models. Here, $N = 399$ is the number of samples ($N_{regions} - 1$ because we exclude self-connections) and $p$ is the number of predictors in the model and is equal to $p = 2$ and $p = 3$ for the one- and two-predictor models (the additional parameter is from the intercept).

**Reporting summary**. Further information on research design is available in the Nature Research Reporting Summary linked to this article.

## Data availability

The raw and minimally-processed HCP and NKI data are available under restricted access to maintain subject privacy. Access to raw and minimally processed data can be obtained by digitally signing a data use agreement (https://db.humanconnectome.org/app/template/Login.vm and http://fcon_1000.projects.nitrc.org/indi/enhanced/neurodata.html). The derivative data generated in this study are provided in the Source Data file. Any additional unrestricted materials are available upon request from the authors. Source data are provided with this paper.

## Code availability

Code for estimating the predictors from structural connectivity and using them to predict functional connectivity is available at https://github.com/brain-networks/local_scfc. Subject specific parcellations were fit with FreeSurfer 6.0.1 using code available here: https://github.com/faskowit/multiAtlasTT and data available here: https://figshare.com/articles/multiAtlasTT_data_hcptrained/7552853. fMRI data were nuisance regressed with code available here: https://github.com/faskowit/app-fmri-2-mat which uses Nilearn's `signal.clean`, from Nilearn 0.5.0.

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

## Acknowledgements

Data were provided [in part] by the Human Connectome Project, WU-Minn Consortium (Principal Investigators: David Van Essen and Kamil Ugurbil; 1U54MH091657) funded by the 16 NIH Institutes and Centers that support the NIH Blueprint for Neuroscience Research; and by the McDonnell Center for Systems Neuroscience at Washington University.

## Author contributions

F.Z.E. and R.F.B. conceived of the study, performed analyses, and generated figures. J.F. contributed and processed all imaging data. F.Z.E., J.F., J.S., B.M., and R.F.B. wrote and edited the manuscript.

## Competing interests

The authors declare no competing interests.
