## [Peer Review File · Nature Communications]

REVIEWER COMMENTS

Reviewer #1 (Remarks to the Author):

This paper investigates the association between functional connectivity and a wide range of metrics of structural connectivity derived from tractography data. Overall this paper is very well written and the quality control and data analyses are thorough. This paper takes a step back compared to previous work where multiple structural metrics are taken together in one multiple regression model, resulting in important and interesting new insights. It demonstrates that predicting FC on a regional level is much more effective than on a whole brain level and that there is considerable regional specificity in the metrics that best predict functional connectivity. This variability is in line with the functional roles of different brain regions. The authors also show which structural metrics complement each other in terms of predicting FC and how the SC-FC coupling differs across the lifespan. I have a number of (mostly minor) comments:

1. The authors investigate many derived structural measures to predict FC. How do these derived measures compare to the R2 that would be achieved by simply using the streamline count matrix as a predictor?
2. The authors claim that even with multiple predictors, the predictability of FC in heteromodal cortices improves little. It would be interesting to fit a multiple regression model with all predictors at the same time to investigate the maximal amount of variance in FC that could be explained by SC in different regions. That would also provide a point of reference for the regional differences shown in figure 3.
3. The variance explained was substantially higher for the Nathan Kline dataset than for HCP. What factors could have caused this difference?
4. The authors show a core-periphery structure in the graph of structural metrics. However, it is not made very clear what it means for a specific metric to be part of the core.
5. In figure 2h, what threshold is used to determine whether regions are coloured or not?
6. The results in figure 1D are interesting but hard to see. A different visualization (like in 1C) would make these findings more interpretable.
7. Are the results in figure 4e and S6 thresholded for statistical significance? And if so, which correction for multiple comparisons was used?
8. In the discussion section the authors suggest that: "These observations suggest that, decentralized patterns of interregional communication may degrade over the human lifespan, prompting a decoupling of functional connectivity from structure.". I wonder what is meant here exactly. Is it that with age, functional connectivity is shaped more strongly by direct (short) connections?
9. In the discussion section I am missing a section on how the pattern of explained variance across the

cortex compares to previous studies using similar approaches (e.g. ref 27, 28)

10. Which gamma value was used in the Louvain clustering algorithm and how was that value selected?

11. Would it be possible to colour the clusters in the figure S3 according to the dominant structural metric in each cluster (making the left out regions grey). The current colour scheme is confusing.

12. Could the age differences between participants explain the observed associations between SC-FC coupling and intelligence?

Reviewer #2 (Remarks to the Author):

This ms investigates three extremely interesting and challenging problems, namely: 1) the link between SC and FC in the sense of prediction, i.e. which predictors are more reliable for determining the FC from the SC; 2) the question about the spatial scale of the SC/FC prediction, i.e. how local or global and how the regionality influence the prediction; and 3) how the SC/FC link changes across the human lifespan.

All three questions are fundamental and were convincingly answered. The authors have demonstrated that global predictions are not reliable and that local predictions (e.g. Euclidean distance and weighted mean first passage time) are very good. Furthermore, the SC/FC coupling favored sensorimotor cortices. They also studied multi-linear models aiming to enhance the level of predictions. They have shown significant improvements using combinations of Euclidean distance and weighted mean first passage time, together with measures of communicability. Finally, when examining changes across lifespan they found that the magnitude of coupling decreased with age, an observation driven by decreases in local coupling of visual and somatomotor systems, whereas SC/FC coupling in higher-order cognitive systems, including the control and default mode networks, went largely unchanged with age and, in some cases, even increased. The results are relevant and solid and therefore I recommend to accept this paper. I have only minor recommendations. I would be more detailed in reviewing the whole-brain literature, and more fair. There were really much more papers than the one that they cited and with more results that makes the decision between which is the best approach for establishing the link between FC and SC more challenging. Many aspects of modelling are extremely useful for addressing the same questions with the addition that they can be used to include also the time dimension in, which is fully neglected by the present approach. So I would really be more balanced in the judging of whole-brain models and stress the complementarity (specially for going not only space but also time prediction information).

Reviewer #3 (Remarks to the Author):

Overall comment:

The authors explored the relationship between structural connectivity (SC) and resting-state functional connectivity (FC) from MRI images collected in human participants. Diffusion-based sparse SC network matrices were transformed using 40 different predictors(models)/parameter-combinations into fully weighted matrices. This allowed the authors to investigate whether different properties of SC predict global and local regional patterns of FC. Similar analyses were applied first in a young adult only dataset (HCP, n=95), and then to a lifespan dataset (NKI, n=542, age=7-85).

This work follows a collection of closely-related papers focused on this topic that some of the authors of the present work have been involved with. Here, the authors included a wide range of testing parameters across multiple models, which is a good practice when performing explorative analysis. The take home is that Euclidean distance- and mean passage time- based models explain the largest amount of variance in FC patterns, an observation which is consistent with prior reports. The regional analysis (e.g., Figure 2) provides interesting evidence that distinct SC-related properties may constrain patterns of FC across different functional networks, although its not clear what establishes these distinctions or why they vary. Additional variance is explained in a subset of locations when including multiple models (but see concerns below), suggesting that there remains an unexplained territory to better understand the complex relationships between structural and functional connectivity. The application of the SC-FC analysis to a broader range of age is novel, although considerable methodological and theoretical challenges constrain interpretation of these results. As a general point, I also felt that many of the analyses were unnecessarily complex and packaged in ways that hindered rather than facilitated understanding (e.g., the core-periphery analysis and associated results, more comments below). In sum, the work adds to prior observations and is largely consistent with previous papers on this topic but now explored in a larger parameter space (e.g., models, regions, age). As mentioned, interpretation of the more novel elements of the present submission are constrained by concerns which are elaborated upon below.

Detailed comments (no particular order):

- How were the 10 class of predictors chosen?
 - o It is unclear how and why the authors focused in on these specific classes of predictors and their subsets of parameters. There are a considerable number of available models, why are the ones included in this manuscript chosen to investigate SC-FC relationships? This should be discussed in the introduction, as it will further educate readers what these models represent, and why they are relevant to specific aspects of SC.
- Greater details and usage of null models should be included
 - o The authors used spin test to general 1000 null models in their regional analysis and supplemental analysis (SC-FC relation with intelligence). A null model should also be used for the global analysis, showing whether the variance explained by the 40 models actually provide better results than a null.
- Clarify one-way ANOVA set-up
 - o In the second section of the results, one-way ANOVAs were conducted to look at main effects of factors and regions. However, it is unclear whether that is based on data that averaged the subjects

together (i.e., what is visualized in Figure 2a), or if a within-subject factor was used to account for subjects. If the ANOVAs were conducted on data where the dimension of subjects are flattened, then the authors should clarify that the “averaged over subject” was not only for visualization purposes only.

- o Is the main effect of regions’ degrees of freedom incorrect? It says (39) even though there are 400 regions (pg. 3).

- Are delta R2 based on adjusted R2?

- o When the authors examined delta R2 based on coupling of predictors, it is not clear that if these were changes in adjusted R2, where the number of predictors in a model is accounted for. If it isn’t, the addition of predictors generally increases R2, but it may not be an actual meaningful increase after accounting for the sheer occurrence of having more variables in the model.

- PCA space used in Figure 2 and 3 are not adequately explained

- o The authors mentioned in the captions of the figures that PCA was used to generate PC1/PC2 to plot the factors contribution. More explanation of what was done is needed, and how those component space should be interpreted.

- o Details are lacking what the edges are in the biplots.

- o Plotting a PCA biplot without the actual axis in its appropriate space makes it difficult to understand the PC space. The angles and distance of a variable with respect to the origin (0,0) has meaning in PC space, and should not be removed.

- Exploration of relationships should focus on using analyses that primarily reveals contributions of variables (e.g., PCA)

- o There are 40 predictors, and the authors tried to explore the relationship between these predictors in relation to the dependent variable (FC). It seems a stretch to force the coupling of predictor, which then results in a count matrix based on their R square changes across participants. Instead, the authors could focus on multivariate analyses such as PCA that conveniently included the information about similarity between predictors in its component space. The count matrix that shows the coupling of predictor forces a relationship that identifies ‘pairs’ of predictors, but in reality, there is no reason that a combination of 3 or more predictors might be even better suited for this problem.

- o Core-periphery analysis is unnecessary for understanding the relation between predictors in which their pair-wise relation is turned into a graph.

- A major take-away is that the SC-FC relationships alter with increasing age, however this conclusion assumes that structural and functional data are comparable in quality across the lifespan.

- o Head motion is strongly associated with resting-state function correlations (e.g., Power et al. 2012; Van Dijk et al., 2012; Ciric et al. 2017), which is critical to account for especially in studies that include participants with heightened motion (e.g., older adults). In the present study, the authors despiked frames that were above a frame displacement of 0.5mm in the lifespan dataset (NKI), which is inadequate in mitigating motion artifacts in the results (see Power et al 2014). Furthermore, no frame-wise correction was performed for the HCP dataset at all. A more stringent motion threshold should be employed in both datasets (HCP and NKI), followed by the reporting of the level of motion artifacts detected and removed. Lastly, the number of contaminated frames should also be controlled for in analysis across subjects. This is critical as motion artifact’s relation to FC is also distance dependent

(Euclidean distance is a major variable in the present manuscript).

o In a similar way, head motion is also associated with SC, even after standard correction for head motion, eddy current, and field distortion (Baum et al. 2018). Head motions impacts the estimated SC in a way that is sensitive to edge consistency and length (which is associated with distance). Furthermore, as white matter degrades with age, and the quality of tractography suffers. This results in poorer estimates of white-matter that would alter the aging effect.

o Given that the quality of both SC and FC is lower in older adults, it is fairly reasonable that their association is also weaker (e.g., increased noise in the data). Applying the current analyses in a lifespan dataset requires a much more rigorous approach that accounts for these known age-related artifacts and challenges in analysis and interpretation of both types of imaging data.

- Floor effect of age-related R2 differences.

o The authors should address the possibility for a floor effect where regions with initially low R2 (Figure 4d) have little room to exhibit age-related decreases in R2. A conjunction map of Fig. 4d and 4e would likely show areas of low R2 showing little decrease or even increases, whereas areas with the highest R2 to begin with has the greatest room to show ‘decreases’ with age. While it is likely that those areas truly exhibit greater age-related decreases in R2, the possibility of a floor effect or even regressing to the mean should be addressed.

Minor:

- Age-related differences observed from a cross-sectional sample should be labeled as ‘differences’ instead of ‘changes’; changes implies within-person changes, which requires longitudinal data.

- Yellow labels or colors are not very visible/readable. Please consider using something like “brown” (e.g., Fig 1c, 2a, 2g)

- Terms that are specific to other types of data (e.g., ‘expression’ from genes, ‘fingerprinting’) creates more confusion than actually help the reader understand the results.

- Due to how similar in setup Figure 1c and 2a are, their y-axis label should indicate that 2a is actually “Mean variance explained across subjects”.

Dear Reviewers,

We wish to thank the editor and all reviewers for their thoughtful comments and suggestions. Our enclosed submission is significantly improved as a result. We provide, here, a short summary of those changes:

1. We compared performance of predictors with that of the raw SC matrix (R1),
2. We tested the effect of combining all predictors into a single multilinear model (R1),
3. Performed statistical testing at regional level for assessing lifespan/intelligence relationships (R1),
4. Included a broader discussion of methodology for linking brain structure and function (R2),
5. Fully replicated all of our results following increasingly stringent motion-correction strategies applied to both datasets (HCP and NKI) (R3), and
6. Addressed statistical concerns, including issues related to global structure-function relationships, the effect of including multiple predictors in the linear model, among others (R3).

Throughout this letter we refer to comments made by the reviewers in *italics*. Changes made to the text are shown in *blue italics* and indented. When appropriate, we highlight the section name where changes were implemented. In addition to this response letter, we include two versions of the revised manuscript: a “marked” version that tracks all of the changes made since the original submission and a “clean” version that includes all changes but without any highlights.

Farnaz Zamani Esfahlani

Rick Betzel

Reviewer #1 (Remarks to the Author):

This paper investigates the association between functional connectivity and a wide range of metrics of structural connectivity derived from tractography data. Overall this paper is very well written and the quality control and data analyses are thorough. This paper takes a step back compared to previous work where multiple structural metrics are taken together in one multiple regression model, resulting in important and interesting new insights. It demonstrates that predicting FC on a regional level is much more effective than on a whole brain level and that there is considerable regional specificity in the metrics that best predict functional connectivity. This variability is in line with the functional roles of different brain regions. The authors also show which structural metrics complement each other in terms of predicting FC and how the SC-FC coupling differs across the lifespan. I have a number of (mostly minor) comments:

1. The authors investigate many derived structural measures to predict FC. How do these derived measures compare to the R² that would be achieved by simply using the streamline count matrix as a predictor?

This is a good point. Assessing the magnitude of structure-function coupling using the streamline matrix would help contextualize the results generated using the derived metrics.

To this end, we performed the following analysis. For each subject and for each brain region, we created a “mask” of its existing connections to other brain regions and extracted their corresponding structural functional connection weights. We then fit a linear model to predict functional connection weights using structural weights. This procedure resulted in a [400 x 95] matrix of whose elements corresponded to variance explained (R² values).

The R² values obtained from the streamline matrices and structural predictors were broadly similar to one another (mean similarity of $r = 0.36 \pm 0.08$). However, there were some important differences.

To better understand how these two approaches compared to one another, we determined for each subject and region, whether its functional connections were better explained by streamline matrices or structural predictors. We found that, for each subject, structural predictors better explained functional connections for 67.5 ± 0.06 regions. Interestingly, the regional preferences exhibited non-random spatial structure. Regions whose functional connections were better explained by the structural predictors included those in control, default mode, dorsal attention, salience, ventral attention, somatomotor, and visual networks. In contrast, streamline matrices better predicted the functional connection weights of the temporo-parietal network.

Collectively, these results suggest that using structural measures to predict functional connection weights improve on predictions made using streamline counts, alone.

We note, however, that there are issues in directly comparing the performance of structural predictors and streamline matrices. Notably, the streamline matrices are sparse (only a fraction of connections exist) whereas the structural predictors are fully weighted (every node pair has some nonzero value). For the sparse matrices, we cannot make predictions about functional connections for which there is no corresponding structural connection.

Fig. S5. Comparison of communication measures and structural connectivity in predicting FC. In the main text we derived a series of communication measures from sparse structural connectivity (SC) data to predict regional FC patterns. Here, we compare the results from the communication measures with the results obtained from using the structural connections directly. In this analysis, we create a “mask” for each region and subject of its structural connections to other regions. We then extract the weights of those structural and functional connections and fit a linear model to explain the functional connection weights in terms of the structural weights. This results in an R^2 value for each region and subject. (a) Variance explained using communication measures averaged across subjects. (b) Variance explained using structural connections alone averaged across subjects. (c) Regional differences in variance explained. Warmer and cooler colors indicate regions whose FC is better predicted using the communication measures and structural connectivity, respectively. (d) Data from panel c but displayed by brain system.

We have now added Fig. S5 to the supplementary material and reference it in the Results section. The revised text reads:

We also repeated this process using structural connection weights instead of derived predictors (see Fig. S5). We found that, while the two techniques generated similar regional patterns of explained variance (mean similarity of $r = 0.36 \pm 0.08$), the predictors generally outperformed structural connectivity, with the FC of $67.5 \pm 0.06\%$ of regions being better explained by predictors derived from the SC matrices than the SC matrix itself.

2. The authors claim that even with multiple predictors, the predictability of FC in heteromodal cortices improves little. It would be interesting to fit a multiple regression model with all predictors at the same

time to investigate the maximal amount of variance in FC that could be explained by SC in different regions. That would also provide a point of reference for the regional differences shown in figure 3.

We again agree with the reviewer that this is an interesting idea and a useful benchmark. We followed the reviewer's suggestion and used all 40 predictors to predict regional FC (see Fig. S8 below). Compared to the single-best predictor, using all predictors resulted in a three-fold increase in R^2 (3.2 +/- 1.2 times greater) (Fig. S8e). Relative to the best two-predictor models, using all predictors still resulted in a greater than two-fold increase in R^2 (2.4 +/- 0.6) (Fig. S8e).

Interestingly, the biggest increases in R^2 as a result of using all predictors included regions in default mode and salience/ventral attention networks. Of the top 10% of regions by increase in R^2 , nearly half came from those two networks, including regions in default mode that exhibited a greater than ten-fold increase (Fig. S8d).

Fig. S8. Predicting FC using all measures. In the main text we predicted regional FC using metrics derived from SC matrices. Specifically, we focused on the variance explained by the best individual predictor (a) and the best pair of predictors (b). Here, we repeat this analysis but using all predictors to predict regional FC. (c) The variance explained at each region after using all predictors. (d) Ratio of variance explained using the best pair of predictors (panel b) and all predictors (panel c). Note that the peak increases fall within default mode (interparietal sulcus; posterior cingulate; precuneus) and salience/ventral attention (anterior cingulate). (e) Boxplot of regional variance explained. (f) Scatterplot of regional variance explained with the single best predictor versus the best pair of predictors (grey) and all predictors (orange). The black line is the identity line.

We now include this figure in the supplementary material. The revised text reads:

For completeness, we all considered the effect of combining all predictors into the same linear model. Note that many of the predictors are correlated, making it difficult to assess their unique contributions. However, we note that including all predictors yields R^2 values that are, on average, 2.4 ± 0.6 times greater than the best pairs of predictors. The regions that benefit the most from the inclusion of all predictors fall within default mode and salience/ventral attention

systems, and, in some cases, exhibit increases in R^2 that are ten times that of the best predictor pair. The results of this additional analysis are summarized in Fig. S8.

3. The variance explained was substantially higher for the Nathan Kline dataset than for HCP. What factors could have caused this difference?

This is an important point and one worth discussing. First, it is reassuring to note that the overall spatial pattern of the variance explained is similar between the two datasets (compare Fig. 2c with Fig. 4d, $r=0.73$), despite the notable differences in the magnitude of overall variance explained. One possible explanation for the difference in maximum magnitudes of these maps is the difference between cohort sizes ($n=95$ versus $n=525$). Since this map represents the maximum R^2 value for each node, it could be the case that in the larger NKI sample, more individual variation in structure-function prediction is captured in the sample. This might lead to specific nodes having a higher maximum R^2 based on a specific individual's differences that would more likely show up in the larger (and more heterogeneous regarding age and community representativeness) NKI sample.

Another possible explanation for the difference in magnitudes is differences in data quality and data preprocessing between the two cohorts. For NKI data, the functional correlations and structural streamline counts were measured relative to a volumetric parcellation in the subject anatomical space (3mm grid for functional scan, 2mm grid for DWI scan, 1mm grid for T1w scan). For HCP, structural streamline counts were measured relative to a volumetric parcellation in the subject anatomical space (0.7mm grid for T1w scan, 1.25 mm grid for DWI scan), but the functional data was provided by the HCP consortium in a standard high-resolution (2mm spacing between vertices) surface space known as fs_LR space. Thus, HCP functional correlations were measured using the same parcellation (Schaefer 400), but in the fs_LR space as opposed to the subject anatomical space. We could speculate that this higher-resolution functional data in the standard space could be a possible explanation for the lower maximum R^2 values in the HCP cohort. It could be the case that the surface functional data, which is thought to better localize functional signals (Coalson et al 2018, PNAS), is less influenced by the Euclidean distance between nodes (which is the top predictor in both datasets, but to a lesser extent in the HCP sample), which could possibly result in the lower maximum R^2 magnitudes. Additionally, a possibility is that the definition of nodes in fs_LR space is slightly different (owing to an alternative tool set used to fit the parcellations) node definitions for the functional and structural brain networks for the HCP data. However, ultimately, the similarity of the spatial pattern of maximum magnitudes reassures us that the effect of this mismatch likely insignificant to the main results of the paper.

We have updated the Discussion section to include this topic. The revised text reads:

Interestingly, while the regional patterns of structure-function coupling in NKI and HCP datasets were correlated, the magnitude of coupling was noticeably stronger. While the origins of this difference remain unclear, possible explanations include differences in sample size ($N = 95$ in HCP versus $N = 542$ in NKI), heterogeneity, and community representativeness. Other possibilities include differences in data quality and preprocessing strategies. In the NKI dataset, functional and structural networks were estimated using a volumetric parcellation in subject anatomical space. In contrast, the HCP data was processed using surface-based analyses. Indeed, recent studies have identified notable differences between these two image processing pipelines (Coalson et al 2018). Future studies should investigate this possibility in greater detail, documenting the effect of different processing pipelines on local structure-function coupling.

4. The authors show a core-periphery structure in the graph of structural metrics. However, it is not made very clear what it means for a specific metric to be part of the core.

We apologize for any confusion and appreciate the opportunity to clarify. In this context, a “core” refers to pairs of predictors (metrics) that frequently appear together in two-term multilinear models. The same core predictors may sometimes be paired with peripheral predictors, but peripheral predictors are infrequently paired together. That is, the core is comprised of predictors that exhibit strong synergies in their ability to predict FC patterns; the periphery is comprised of predictors that exhibit relatively weak synergies.

We have updated the manuscript to better clarify this. The revised text reads:

In this context, a “core” refers to pairs of predictors (metrics) that frequently appear together in two-term multilinear models. The same core predictors may sometimes be paired with peripheral predictors, but peripheral predictors are infrequently paired together. That is, the core is comprised of predictors that exhibit strong synergies in their ability to predict FC patterns; the periphery is comprised of predictors that exhibit relatively weak synergies.

5. In figure 2h, what threshold is used to determine whether regions are coloured or not?

We apologize for any confusion. In this figure, the colors of brain regions correspond to the fraction of subjects for which a given predictor was optimal, i.e. yielded the greatest R^2 value. The underlying values are continuous and range between zero – a given predictor was optimal for no subjects – up to one – a given predictor was optimal for all subjects. The colors, therefore, are also continuous, with gray representing a value of zero and brighter colors indicating increasing fractions of subjects. For the sake of visualization, we saturated the colormap at the fraction $3/10$, so that if any regions above a value of 0.3 are assigned the brightest possible color.

We have updated the figure caption to indicate this more clearly. The revised caption reads:

Note here that the colors projected onto the surface are continuous; grays correspond to regions where a predictor was optimal for few subjects while brighter colors correspond to regions where a predictor was optimal for many subjects. In all panels the colorscale was capped at a value of 0.3.

6. The results in figure 1D are interesting but hard to see. A different visualization (like in 1C) would make these findings more interpretable.

We appreciate the suggestion and have updated Figure 1D. See below. In this panel, the first row represents variance explained using only direct connections with path length of one ($PL = 1$). The next two rows show variance explained at path lengths two and three ($PL = 2$ and $PL = 3$).

7. Are the results in figure 4e and S6 thresholded for statistical significance? And if so, which correction for multiple comparisons was used?

This is an important point. We note that, in both cases, we did not test for significance at the region level, but at the level of brain systems (Figures 4e and S6 were accompanied by boxplots showing this). We agree, however, that it would be useful to also test for statistical significance at the regional level. In both cases, we adjusted our critical p-value to accommodate a 5% false discovery rate (Benjamini & Hochberg, 1995). In the case of the age correlation (shown in Figure 4e), we found robust negative correlations involving regions in the somatomotor and visual systems ($p_{\text{adj}} = 0.018$). In the case of the intelligence correlation, no regions passed the multiple comparisons correction. See Fig. S11, below.

Fig. S11. Statistical thresholding of regional correlation maps. In the main text we computed the correlation of age and intelligence with the regional measures of structure-function coupling (R^2). In those analyses, we performed statistical testing at the level of brain systems. That is, we identified systems whose average correlation was stronger – more positive or negative – than expected by chance. Here, we perform statistical testing at a regional level, correcting for multiple comparisons by adjusting the critical value to accommodate a false discovery rate of $q = 0.05$, i.e. 5%. In panel a, we show results of the age and R^2 correlation ($p_{\text{adj}} = 0.018$). We note that no regions survive corrections for multiple comparisons in the case of correlation with intelligence.

We have now included a thresholded brain surface plot in the Supplementary Material (Fig. S11) and have made edits to the text. The revised text reads:

We include thresholded surface maps of the correlations between R^2 with both age and intelligence in the supplementary material (Fig. S11).

8. *In the discussion section the authors suggest that: “These observations suggest that, decentralized patterns of interregional communication may degrade over the human lifespan, prompting a decoupling of functional connectivity from structure.” I wonder what is meant here exactly. Is it that with age, functional connectivity is shaped more strongly by direct (short) connections?*

We appreciate the opportunity to clarify. One of the principal findings in our study is that the whole-brain correlation between FC and predictors decreases with age (Fig. 4a). In parallel, we observed that the mean first passage time metric (MFPT) also becomes less frequent with increasing age. In the discussion, we speculated that these two observations might be related. That is, the decrease in coupling could be explained by the decreased use of MFPT as a communication policy.

We have updated the text to better clarify this assertion. The revised text reads:

Based on these findings and along with the observation that the magnitude of global structure-function coupling decreases with age, we speculate that, decentralized patterns of interregional communication may degrade over the human lifespan, prompting a decoupling of functional connectivity from structure.

9. *In the discussion section I am missing a section on how the pattern of explained variance across the cortex compares to previous studies using similar approaches (e.g. ref 27, 28)*

This is a good point and we regret the omission. We have updated the Discussion section to contextualize, more broadly, the links between structure and function from a local perspective. The new text reads:

Here, we extend the matrix-based prediction approach from a global, whole-brain level to a the level of individual brain regions. Our approach builds upon other recent studies that examined heterogeneity in coupling patterns across regions (Baum et al 2020; Vazquez et al 2019) and systems (Osmanliouglu 2019). Despite differences in dataset, parcellation, and choice of predictor, our results are closely aligned with those of previous studies, which reported strong coupling in sensorimotor cortices and weaker coupling in heteromodal regions. Interestingly, heteromodal cortex includes regions that expanded dramatically over the course of mammalian evolution (Buckner 2013) and others that develop at a protracted rate (Hill 2010), leading to the hypothesis that their interregional connectivity may be shaped by different organizational principles than regions in unimodal cortex and serving as a possible explanation for the weakened structure-function coupling

10. *Which gamma value was used in the Louvain clustering algorithm and how was that value selected?*

We regret the omission of this detail and are happy to clarify. The reviewer is referring to the resolution parameter, gamma, that appears in the modularity equation. Changing the value of gamma changes the scale of the detected communities, i.e. the number and size of clusters.

Here, we use the uniform null model in our modularity equation. This null model is especially well-suited for analysis of correlation networks (see Bazzi et al 2016). Given this null model, our modularity matrix is defined as: $B_{ij} = S_{ij} - \gamma P_{ij}$, where P_{ij} is equal to 1 for all $\{i, j\}$ and where γ is the resolution parameter.

Although, in principle, the resolution parameter could be varied to detect different-sized communities, we performed community detection with its value fixed at $\gamma = \langle S \rangle = 0.296 \approx 0.3$. That is, the resolution parameter was set to mean similarity across all pairs of regions. Our rationale for doing so was that we wanted to identify communities of regions whose mutual similarity to one another exceeded a “typical” value.

For completeness, we reanalyzed these data while varying the γ parameter and, in addition to the communities reported in the original submission, now show examples of communities at other γ values. The panel from the revised figure is shown below:

In addition, we updated our description of the community detection procedure. It now reads:

where $B_{ij} = S_{ij} - \gamma P_{ij}$. In this expression, P_{ij} is the expected weight of the connection between regions i and j and γ is a structural resolution parameter that tunes the number of size of detected communities. For simplicity, we set $\gamma = \langle S \rangle = 0.296 \approx 0.3$ and used this value for all pairs of brain regions. For completeness, however, we also tested other resolution parameter values, ranging from $\gamma = 0$ to $\gamma = 1$ in increments of 0.025. We show some of these communities in Fig. S3d.

11. *Would it be possible to colour the clusters in the figure S3 according to the dominant structural metric in each cluster (making the left out regions grey). The current colour scheme is confusing.*

We agree that this type of visualization is useful and have adopted it for communities shown in Figure S3.

12. *Could the age differences between participants explain the observed associations between SC-FC coupling and intelligence?*

This is an interesting hypothesis and one that could be tested by regressing age from the intelligence measures and, using the residuals, computing their correlation with variance explained. We performed this procedure and found that the regional correlation pattern – a [400 x 1] vector – was almost identical to the correlation pattern obtained without regressing out age ($r=0.986$). These results suggest that differences in age are likely not drivers of the correlation between SC-FC coupling and intelligence.

Reviewer #2 (Remarks to the Author):

This ms investigates three extremely interesting and challenging problems, namely: 1) the link between SC and FC in the sense of prediction, i.e. which predictors are more reliable for determining the FC from the SC; 2) the question about the spatial scale of the SC/FC prediction, i.e. how local or global and how the regionality influence the prediction; and 3) how the SC/FC link changes across the human lifespan.

All three questions are fundamental and were convincingly answered. The authors have demonstrated that global predictions are not reliable and that local predictions (e.g. Euclidean distance and weighted mean first passage time) are very good. Furthermore, the SC/FC coupling favored sensorimotor cortices. They also studied multi-linear models aiming to enhance the level of predictions. They have shown significant improvements using combinations of Euclidean distance and weighted mean first passage time, together with measures of communicability. Finally, when examining changes across lifespan they found that the magnitude of coupling decreased with age, an observation driven by decreases in local coupling of visual and somatomotor systems, whereas SC/FC coupling in higher-order cognitive systems, including the control and default mode networks, went largely unchanged with age and, in some cases, even increased. The results are relevant and solid and therefore I recommend to accept this paper.

I have only minor recommendations. I would be more detailed in reviewing the whole-brain literature, and more fair. There were really much more papers than the one that they cited and with more results that makes the decision between which is the best approach for establishing the link between FC and SC more challenging. Many aspects of modelling are extremely useful for addressing the same questions with the addition that they can be used to include also the time dimension in, which is fully neglected by the present approach. So I would really be more balanced in the judging of whole-brain models and stress the complementarity (specially for going not only space but also time prediction information).

We thank the reviewer for their positive comments. We also agree that a more balanced discussion of the literature on modeling the link between FC and SC would broaden the appeal of our submission.

To this end, we have revised our Discussion section, adding two new paragraphs:

1. The first addresses the absence of temporal information in matrix-based predictors of FC.
2. The second discusses tradeoffs between different methods for studying structure-function relationships, emphasizing that they can and should be viewed as complementary to one another.

Additionally, in line with the reviewer's suggestion, we added additional references to the discussion of global structure-function coupling.

This new section reads:

Additionally, matrix-based predictors fundamentally lack a temporal dimension. That is, they compress information about dynamical processes, e.g. diffusion, navigation, shortest paths routing, into matrix form, the precise temporal evolution of those processes is lost. In contrast, biophysical models document the temporal evolution of activity, generating spike trains, voltage traces, or hemodynamic signals from cells, populations, or regions (Deco 2008, Deco 2011, Ritter 2013). These temporal data provide an additional target for modeling studies; rather than simply matching the correlation structure of brain activity, can a model also replicate its time-varying

features? Indeed, while recent work has begun to investigate time-varying structure-function coupling (Kong 2021, Liu 2021, Liegeois 2016, Fukushima 2018), future studies are necessary.

In summary, there exists a spectrum of methods for assessing and modeling structure-function coupling in empirical data (Honey 2010, Sanz Leon 2015, Sanz Leon 2013, Batista 2018, Sarwar 2021, Lynn 2019, Becker 2018, Rosenthal 2018). Each method possesses a distinct set of advantages and disadvantages, such that some research questions are better addressed by one method and not another. For instance, biophysical models offer a neurobiological account of the relationships between structure and function, while making population-level time courses available for analysis. On the other hand, the complexity of biophysical models might make them unappealing for studies where the aim is to generate a biomarker of structure-function coupling. In that case, a simple correlation between SC and FC weights may suffice. In general, these methods should be viewed as complementary rather than adversarial, affording researchers the opportunity to examine data under different sets of assumptions, across spatial scales, with varying levels of neurobiological realism, and using different computational tools.

Reviewer #3 (Remarks to the Author):

Overall comment:

The authors explored the relationship between structural connectivity (SC) and resting-state functional connectivity (FC) from MRI images collected in human participants. Diffusion-based sparse SC network matrices were transformed using 40 different predictors(models)/parameter-combinations into fully weighted matrices. This allowed the authors to investigate whether different properties of SC predict global and local regional patterns of FC. Similar analyses were applied first in a young adult only dataset (HCP, n=95), and then to a lifespan dataset (NKI, n=542, age=7-85).

This work follows a collection of closely-related papers focused on this topic that some of the authors of the present work have been involved with. Here, the authors included a wide range of testing parameters across multiple models, which is a good practice when performing explorative analysis. The take home is that Euclidean distance- and mean passage time- based models explain the largest amount of variance in FC patterns, an observation which is consistent with prior reports. The regional analysis (e.g., Figure 2) provides interesting evidence that distinct SC-related properties may constrain patterns of FC across different functional networks, although its not clear what establishes these distinctions or why they vary. Additional variance is explained in a subset of locations when including multiple models (but see concerns below), suggesting that there remains an unexplained territory to better understand the complex relationships between structural and functional connectivity. The application of the SC-FC analysis to a broader range of age is novel, although considerable methodological and theoretical challenges constrain interpretation of these results. As a general point, I also felt that many of the analyses were unnecessarily complex and packaged in ways that hindered rather than facilitated understanding (e.g., the core-periphery analysis and associated results, more comments below). In sum, the work adds to prior observations and is largely consistent with previous papers on this topic but now explored in a larger parameter space (e.g., models, regions, age). As mentioned, interpretation of the more novel elements of the present submission are constrained by concerns which are elaborated upon below.

Detailed comments (no particular order):

- *How were the 10 class of predictors chosen?*

o It is unclear how and why the authors focused in on these specific classes of predictors and their subsets of parameters. There are a considerable number of available models, why are the ones included in this manuscript chosen to investigate SC-FC relationships? This should be discussed in the introduction, as it will further educate readers what these models represent, and why they are relevant to specific aspects of SC.

This is an important point, and we appreciate the opportunity to elaborate. Our study builds on growing body of work in which sparse structure connectivity matrices are transformed into fully weighted matrices. The interregional weights in these matrices correspond to the capacity for two regions to communicate under some “communication policy.” For instance, in a shortest paths matrix, the weights encode the smallest number of hops (or if the network is weighted, the minimum cost) of going from a source node to a target node. If shortest paths are the primary means by which brain regions communicate with one another, then might expect that the weights of the shortest path matrix be correlated with the weights of functional connections.

Shortest paths represent an example of a “centralized” communication policy in that using a shortest path for signaling requires complete knowledge of a network’s global topology – a particle (signal) moving from a source region to a target region needs to “know” which nodes are on the shortest path and which connections need to be followed in order to stay on that path.

In contrast, a number of studies have proposed “decentralized” communication policies. These include diffusion processes (random walks) and network navigation – where a particle moves from one node to another according to some greedy policy, e.g. choose the connected neighbor nearest the target in some metric space.

In selecting predictors, our aim was to sample this space and include predictors based on both centralized and decentralized processes and, when those predictors are parameterized, at a range of parameter values. In addition, we included other similarity-based predictors, e.g. matching index, cosine similarity, etc., and Euclidean distance. These measures, which do not fit neatly along the centralized/decentralized communication spectrum, were included because previous studies have identified topological similarity and the geometry of the space in which brains are embedded as key features that shape the correlation structure of their activity. We note that code for implementing these models is made public *via* our lab’s github repository, allowing users the freedom to change parameter values or test additional predictors not included in the manuscript (https://github.com/brain-networks/local_scfc).

We have updated the introduction of the manuscript to better to clarify why we selected the predictors we did. The revised text reads:

Broadly, the space of communication models can be organized along a single axis, with models positioned according to how “centralized” or “decentralized” their communication policy is (Avena-Koenigsberger et al 2019; Avena-Koenigsberger et al 2018). Shortest paths, for instance, are an example of a centralized communication policy in that using a shortest path for signaling requires complete knowledge of a network’s global topology. That is, a particle (signal) moving from a source region to a target region needs to “know” which nodes are on the shortest path and which connections need to be traversed to stay on that path. In contrast, “decentralized” communication policies require no knowledge and include diffusion processes (random walks) (Messe et al 2015) and network navigation (Boguna et al 2009) – where a particle moves from one node to another according to some greedy policy, e.g. choose the connected neighbor nearest the target in some metric space. Alongside centralized and decentralized models are similarity- (Hilgetag et al 2000) and distance-based measures (Betzel et al 2019), which assess the overlap of brain regions’ connectivity patterns or their nearness to one another in space.

...

Here, we address these limitations directly. Using data from the Human Connectome Project (van Essen et al 2013) and a suite of communication models (predictors) based on both centralized and decentralized processes, topological similarity, and spatial embedding, we show that at the single-subject level, communication models fit at the regional level outperform those fit globally.

- *Greater details and usage of null models should be included*

o The authors used spin test to general 1000 null models in their regional analysis and supplemental analysis (SC-FC relation with intelligence). A null model should also be used for the global analysis, showing whether the variance explained by the 40 models actually provide better results than a null.

This is an important point. To address this concern, we calculated the observed structure-function coupling magnitude (R^2) for each subject and predictor. We also obtained 1000 null values of the same measure using the spin test procedure described in the main text. Using these null values, we z-scored the observed R^2 measure.

This procedure results in 95 z-scores for each of the 40 predictors (see Fig S1). We evaluated these data in two ways. First, we transformed the z-scores into p-values and calculated for each predictor the fraction of subjects that exhibited a statistically significant correlation (corrected for multiple comparisons by fixing the false discovery rate at 5% and adjusting the critical p-value; $p_{\text{adjusted}} < 0.0208$). We found that 100% of subjects exhibited significant structure-function correlations in 21/40 predictors and that at least 90% in 30/40. Notably, fewer than 5% of subjects were statistically significant in three predictors: the binary flow graph at a Markov time of $t = 10.0$, binary communicability, and weighted path length with a weight-to-cost transformation of -4 .

In addition, we identified predictors whose distribution of z-scores fully excluded a value of 0. The results using this method largely converge with those of the first: 75% of predictors fully exclude a value of zero. The remaining 25% include predictors identified using the first method and parametrically similar predictors along with some of the path transitivity measures.

In general, these results support the hypothesis that, although the variance explained by global structure-function coupling is weak relative to the local coupling, for most predictors and most subjects, the correlations exceed statistical significance.

We now include this figure (Fig. S1) in the Supplementary Material and reference this analysis in the main text. The revised text reads:

We also tested the statistical significance of the variance explained using a spatially-constrained permutation model (Markello 2021) (see Fig S1).

Fig. S1. Statistical analysis of global structure-function coupling. In the main text we reported structure-function coupling at the global (whole-brain) level. Here, we use spin tests to assess the statistical significance of the variance explained. Specifically, we calculated the observed structure-function coupling magnitude (R^2) for each subject and predictor. We also obtained 1000 null values of the same measure using the spin test procedure described in the main text. Using these null values, we z-scored each observed R^2 measure. This procedure results in 95 z-scores for each of the 40 predictors. We evaluated these data in two ways. First, we transformed the z-scores into p-values and calculated for each predictor the fraction of subjects that exhibited a statistically significant correlation (corrected for multiple comparisons by fixing the false discovery rate at 5% and adjusting the critical p-value; $p_{\text{adj}} = 0.0208$). In addition, we identified predictors whose distribution of z-scores fully excluded a value of 0. In this figure, we show for each predictor the distribution of z-scores. Gray rectangles around predictors identify those whose distribution included a value of zero and were considered "not significant". The row of numbers below the boxplot indicates, for each predictor, the fraction of subjects that exhibited a statistically significant structure-function correlation.

- Clarify one-way ANOVA set-up

o In the second section of the results, one-way ANOVAs were conducted to look at main effects of factors and regions. However, it is unclear whether that is based on data that averaged the subjects together (i.e., what is visualized in Figure 2a), or if a within-subject factor was used to account for subjects. If the ANOVAs were conducted on data where the dimension of subjects are flattened, then the authors should clarify that the "averaged over subject" was not only for visualization purposes only.

o Is the main effect of regions' degrees of freedom incorrect? It says (39) even though there are 400 regions (pg. 3).

We appreciate the reviewer bringing this to our attention. When inspecting the reported ANOVA results, we noticed two errors in reporting. First, the test statistic for the ANOVA of region-averaged R^2 values was copied from the previous section and incorrectly listed as $F(39) = 326.6$. The correct test statistic is $F(39) = 141.6$. Second, when checking the ANOVA of subject-averaged R^2 values across regions, we realized that the test had been performed over the wrong dimension (predictors) when in fact we intended to perform it across regions.

We apologize for this error and thank the reviewer for discovering it. In the revised manuscript we have corrected both issues and now report the correct results and degrees of freedom. Additionally, we now clarify that the subject-averaged results are used not only for visualization purposes but also for statistical analysis.

The revised text reads:

To visualize these results and for subsequent statistical analyses, we averaged over subjects and plotted the mean variance explained for each region and predictor (Fig 2a). As in the previous section, we found considerable variability across predictors (one-way ANOVA R^2 ; $F(39) = 141.6$; $p < 10^{-15}$) but also across regions (one-way ANOVA R^2 ; $F(399) = 35.4$; $p < 10^{-15}$), confirming that both regions and predictors differ from one another in terms of their mean variance explained.

- Are delta R^2 based on adjusted R^2 ?

o When the authors examined delta R^2 based on coupling of predictors, it is not clear that if these were changes in adjusted R^2 , where the number of predictors in a model is accounted for. If it isn't, the

addition of predictors generally increases R^2 , but it may not be an actual meaningful increase after accounting for the sheer occurrence of having more variables in the model.

This is an important point. We repeated our analyses by calculating the adjusted R^2 for both the one- and two-term models. Specifically, we calculated the adjusted R^2 as:

$$R_{adj}^2 = 1 - \frac{(1 - R^2)(N - 1)}{N - p - 1}$$

where $N = 399$ is the number of samples (we exclude self-connections for all regions) and p is the number of parameters and is equal to 2 and 3 for the one- and two-term models (the additional parameter is due to the intercept in both models).

In general, we find virtually no difference in the spatial pattern of ΔR^2 when we use the adjusted version compared to the unadjusted version (similarity with the unadjusted version is $r = 0.99$, $p < 10^{-15}$; see Fig. S7 below).

We now include the figure (Fig. S7) below figure in the supplementary material and reference it in-text. The revised text reads:

In the supplementary material we repeat this analysis using the adjusted R^2 to confirm that changes in variance explained are not simply a consequence of the additional parameter in the linear regression model (Fig.S 6).

Fig. S7. Regional improvement in adjusted R^2 . In the main text we reported the change in variance explained, ΔR^2 as a result of including two predictors. In general, we might expect increases in R^2 simply due to the inclusion of a second parameter. Here, we show an analogous plot using the adjusted R^2 measure, which takes into account the number predictors. Specifically, we calculated $R_{adjusted}^2 = 1 - (1 - R^2)(N-1)/(N - p - 1)$ for each region for both the one- and two-term models. Here, $N = 399$ is the number of samples ($N_{regions} - 1$ because we exclude self-connections) and p is the number of predictors in the model and is equal to $p = 2$ and $p = 3$ for the one- and two-predictor models (the additional parameter is from the intercept). In general, we find excellent correspondence between ΔR^2 and $\Delta R_{adjusted}^2$ ($r = 0.99$, $p < 10^{-15}$). These observations suggest that the change in variance explained is not likely driven simply by an increase in the number of parameters.

- PCA space used in Figure 2 and 3 are not adequately explained
o The authors mentioned in the captions of the figures that PCA was used to generate PC1/PC2 to plot the factors contribution. More explanation of what was done is needed, and how those component space should be interpreted.

o Details are lacking what the edges are in the biplots.

o Plotting a PCA biplot without the actual axis in its appropriate space makes it difficult to understand the PC space. The angles and distance of a variable with respect to the origin (0,0) has meaning in PC space, and should not be removed.

We apologize for the omission of details about the PCA plots. In line with the reviewer's suggestions, have done the following:

1. Updated the figure caption to include a more detailed description of the PCA analysis.
2. Updated the figure caption to indicate what edges represent.
3. Included in both PCA plots the axis.

We also note that this PCA analysis was performed largely as a means of embedding predictors in a low-dimensional space and, other than helping to reveal similarities between different predictors, we do not analyze the results further.

The revised figure captions read:

1. For Figure 2:

(b) Spatial embedding of predictors. The coordinates were determined using the following procedure: 1) calculating the similarity of regional correlations for every pair of predictors, 2) thresholding this matrix to retain the k=4 nearest neighbors for each predictor, and 3) performing a principal component analysis of the thresholded and symmetrized matrix. Coordinates represent the first two principal components, PC1 and PC2. Broadly, predictors that are near/distant to one another in principal component space exhibit similar/dissimilar patterns of regional correlations. In this plot, edges exist between predictors if they are nearest neighbors.

2. For Figure 3:

In this plot, coordinates were determined by: 1) thresholding the count matrix to retain, for each predictor, its k = 4 nearest neighbors, and 2) performing a principal component analysis on the thresholded and symmetrized matrix. Here, the coordinates represent the first two principal components, PC1 and PC2. Predictors that are near/distant from one another in principal components space pair with similar/dissimilar sets of predictors when improving R². Predictors are joined by an edge if they are considered nearest neighbors.

• *Exploration of relationships should focus on using analyses that primarily reveals contributions of variables (e.g., PCA)*

o There are 40 predictors, and the authors tried to explore the relationship between these predictors in relation to the dependent variable (FC). It seems a stretch to force the coupling of predictor, which then results in a count matrix based on their R square changes across participants. Instead, the authors could focus on multivariate analyses such as PCA that conveniently included the information about similarity between predictors in its component space. The count matrix that shows the coupling of predictor forces a relationship that identifies 'pairs' of predictors, but in reality, there is no reason that a combination of 3 or more predictors might be even better suited for this problem.

o Core-periphery analysis is unnecessary for understanding the relation between predictors in which their pair-wise relation is turned into a graph.

This is an important point, and we appreciate the opportunity to clarify decisions made as part of our analysis pipeline. In general, we agree with the reviewer that there are multiple strategies for exploring structure-function coupling. While we adopted an approach using single or pairs of predictors, deriving principal components from correlated predictors is another. Here, we explore this type of analysis and its utility for predicting function from structure. We describe the results of this analysis below and report the same findings in the Supplementary Material.

Specifically, we performed the following steps. For each subject, we vectorized and concatenated their 40 predictors (Fig. S9a) into a [NumEdges x 40] matrix and z-scored its columns (Fig. S9b). This matrix was used as input for the PCA algorithm, which generated 40 orthonormal components, the loading of each predictor onto each component, and the variance explained by those components (Fig. S9c). This procedure was performed separately for each subject.

We focus on the top 5 PCs, as they collectively account for >75% variance in all subjects (Fig. S9d). Across subjects, the PCs explained $p_1 = 52.7 \pm 1.3\%$, $p_2 = 9.7 \pm 0.5\%$, $p_3 = 6.5 \pm 0.4\%$, $p_4 = 5.3 \pm 0.2\%$ and $p_5 = 4.6 \pm 0.2\%$ variance, respectively (here, subscripts denote PC number).

The [NumEdges x 1] components themselves were highly correlated, with mean absolute correlations of $r_1 = 0.87 \pm 0.02$, $r_2 = 0.74 \pm 0.03$, $r_3 = 0.57 \pm 0.05$, $r_4 = 0.34 \pm 0.12$, and $r_5 = 0.44 \pm 0.15$, suggesting that, even without aligning the components, there was a broad correspondence across individuals, especially in the case of PC1 and PC2 (Fig. S9e). We show group-averaged components in Fig. S9g.

Fig. S9. Predicting FC using all PCs. In the main text we predicted regional patterns of FC using a series of predictors and later identified pairs of predictors that maximally improved these predictions. However, because predictors are correlated with one another, interpreting the results of this analysis may be difficult. Here, we use principal component analysis (PCA) to generate an orthonormal basis set from the predictors and use the resulting principal components (PCs) to predict FC. We performed PCA at the subject level using the 40 predictor matrices as input (a). Each matrix was (b) vectorized

and z-scored before being decomposed into (c) principal components, coefficients, and singular values. (d) Here, we focus on the top five PCs, which accounted for >75% of variance across all subjects. We found that the top five PCs explain 52.7 ± 1.3 , 9.7 ± 0.5 , 6.5 ± 0.4 , 5.3 ± 0.2 and 4.6 ± 0.2 percent variance, respectively. Despite PCA being performed on individual subjects, the PCs and coefficients were highly correlated across individuals. (e) We found that the similarity of the top five PCs across subjects was 0.87, 0.74, 0.57, 0.31, and 0.42, respectively. This is despite the fact that the dimensionality of each PC was $[79800 \times 1]$. (f) To similarity of coefficients from the top five PCs was 0.99, 0.98, 0.95, 0.57, and 0.79, respectively. (g) Group-averaged matrix representations of the first five PCs. We used PCs to make predictions about FC, fitting one linear model per region, per PC, and per subject. We identified for each subject the PC that yielded the greatest R^2 and averaged R^2 values across subjects. In panels, h and i we show R^2 values and the optimal PC projected onto cortical surface. We compared R^2 values reported in the main text with those generated using the PCA-based approach. We find that results from the main text outperform those generated using PCs (paired-sample t-test, $p < 10^{-15}$). We present this comparison in panel j. As in the main text, we also tested the effect of combining pairs of predictors in the same multilinear model. Again, we found that the two-predictor models reported in the main text outperform the PCA approach (paired sample t-test, $p < 10^{-15}$). A boxplot highlighting this comparison is shown in panel k. (l) Increases in R^2 as a results of including an additional PC as a predictor. Interestingly, the combinations of predictors that lead to the greatest improvement tended to involve PC1 pairing with other components. (k) A matrix plot of how frequently pairs of predictors were identified as the optimal model.

The $[40 \times 1]$ loadings of predictors onto the PCs were also highly correlated across subjects: $c_1 = 0.99 \pm 0.001$, $c_2 = 0.98 \pm 0.01$, $c_3 = 0.95 \pm 0.03$, $c_4 = 0.57 \pm 0.25$, $c_5 = 0.79 \pm 0.18$, again suggesting a conserved set of components across subjects (Fig. S9f).

Next, we used PCs to generate predictions of regional FC. We followed the same analysis pipeline as the main text. Instead of fitting 40 linear models (one for each communication measure), we fit 5 models (one for each PC). For each region and each subject, we identified the PC with the greatest R^2 and then averaged this value across subjects, yielding a single $[400 \times 1]$ vector (Fig. S9h). We compared this vector against the one reported in the main text (Fig. 2c) and found excellent correspondence ($r = 0.96$). However, the values estimated using PCA were significantly less than those estimated using the predictors (paired sample t-test, $p < 10^{-15}$; Fig. S9j), indicating that while the patterns are similar, the individual predictors actually outperform the PCA approach. We also find that for most regions (59.3%) PC1 is the predictor that explains the greatest amount of variance (Fig. S9i).

Next, we tested whether combinations (pairs) of PCs could lead to improvements in R^2 . In general, we found that the optimal combinations were PC1+PC2 and PC1+PC3 (Fig. S9k). As in the main text, the regions with the greatest increases in R^2 were concentrated in sensorimotor systems (Fig. S9l). However, as in the previous section, we find that the R^2 values of the multi-linear models with PCs were significantly less than those reported in the main text (paired sample t-test, $p < 10^{-15}$; Fig. S9k).

In general, these results corroborate those reported in the main text. Specifically, they support the hypothesis that local structure-function coupling is strongest in sensorimotor systems and that combinations of predictors – this time pairs of PCs – yield improvements in variance explained.

We now include this figure (Fig. S9) and results results in the Supplementary Material. The revised text reads:

In parallel, we repeated the above analysis using principal components derived from predictors rather than the predictors themselves. This procedure helps address concerns related to correlated predictors. In general, the results of this analysis converge with those reported here. See Fig. S9 for a summary of these results.

- *A major take-away is that the SC-FC relationships alter with increasing age, however this conclusion assumes that structural and functional data are comparable in quality across the lifespan.*
 - o *Head motion is strongly associated with resting-state function correlations (e.g., Power et al. 2012; Van Dijk et al., 2012; Ciric et al. 2017), which is critical to account for especially in studies that include participants with heightened motion (e.g., older adults). In the present study, the authors despiked frames that were above a frame displacement of 0.5mm in the lifespan dataset (NKI), which is inadequate in mitigating motion artifacts in the results (see Power et al 2014). Furthermore, no frame-wise correction was performed for the HCP dataset at all. A more stringent motion threshold should be employed in both datasets (HCP and NKI), followed by the reporting of the level of motion artifacts detected and removed. Lastly, the number of contaminated frames should also be controlled for in analysis across subjects. This is critical as motion artifact's relation to FC is also distance dependent (Euclidean distance is a major variable in the present manuscript).*
 - o *In a similar way, head motion is also associated with SC, even after standard correction for head motion, eddy current, and field distortion (Baum et al. 2018). Head motions impacts the estimated SC in a way that is sensitive to edge consistency and length (which is associated with distance). Furthermore, as white matter degrades with age, and the quality of tractography suffers. This results in poorer estimates of white-matter that would alter the aging effect.*
 - o *Given that the quality of both SC and FC is lower in older adults, it is fairly reasonable that their association is also weaker (e.g., increased noise in the data). Applying the current analyses in a lifespan dataset requires a much more rigorous approach that accounts for these known age-related artifacts and challenges in analysis and interpretation of both types of imaging data.*

We agree with the reviewer that the effects of in-scanner motion could be better addressed. To this end, we performed a series of additional analyses and have also updated our Discussion section to candidly discuss possible limitations.

First, we reanalyzed HCP data employing a more rigorous motion threshold. Specifically, following preprocessing (which was left unchanged), we:

1. censored any frames with relative motion > 0.15,
2. censored frames that were within 2 or fewer frames of any frame dropped in step 1, and
3. retained only contiguous sequences of frames of length 5.
4. In addition, we completely excluded any scans that contained fewer than 50% of usable frames following steps 1-3. Any subject dropped a scan based on this criterion was also removed from all subsequent analyses.

Based on these criteria, we retained 70/95 subjects. Of the remaining 70 subjects, they retained, on average, $90.3 \pm 0.97\%$ of their frames. These retained frames corresponded to an average relative movement of 0.067 ± 0.01 . In contrast, the censored frames had average relative movement of 0.15 ± 0.03 (Note: these values include frames that are sub-threshold in terms of motion but occur in proximity to high-motion frames or represent short sequences of low-motion frames).

Using these data, we calculated FC matrices for each subject. We then replicated the principal results of our study. Specifically, we found:

- 1) We used the 40 predictors to predict each subjects FC. This procedure resulted in a [40 predictor x 70 subject] matrix of R^2 values, which we compared with the results reported in the manuscript. To calculate the similarity of the new, motion-corrected results with those in the main text, we transformed the [40 x 70] matrices into [2800 x 1] vectors and computed their similarity (Pearson correlation). We found a strong correspondence ($r = 0.997$). These findings held at the subject level (calculating column-wise similarity; $r = 0.999 \pm 1.3 \times 10^{-4}$).
- 2) Next, we performed an analogous procedure at the regional level, resulting in a [400 region x 40 predictor x 70 subject] matrix of R^2 values. Again, we assessed the similarity of motion-corrected results with those in the main text by vectorizing each matrix, transforming them into vectors of length [1,120,000 x 1] and computing their Pearson correlation. As before, we found an excellent correspondence ($r = 0.995$). We also repeated this analysis at the subject level by transforming each of the [400 x 40] matrices into a [16,000 x 1] vector and computing, for each subject, the similarity between the originally reported and motion corrected R^2 values. Once again, we found a strong correspondence $r = 0.9995 \pm 6.13 \times 10^{-4}$.
- 3) Finally, we modeled FC using pairs of predictors, identified for each region the optimal pair (the one that explained the most variance), and summarized these results with a “count” matrix, as in the main text. We then computed the correlation of the motion corrected count matrix (its upper triangle) with the matrix reported in the original submission. We found a strong correlation of $r = 0.997$.

Collectively, these results suggest that none of the effects reported in the main text using the HCP data are obviously confounded by in-scanner motion.

We now report these results, including the motion statistics, in the main text and include a supplementary figure showing the scatterplots. The revised text reads:

In the main text, we analyzed data that had been processed using the above procedure. We also performed extensive post-processing of these data to reduce the likelihood that in-scanner motion contributed to any reported effects (Power 2012, Power 2014, Ciric 2017, Parkes 2018). Specifically, we implemented the following steps for the HCP data:

1. *Using the Movement_RelativeRMS.txt time series, we identified frames with motion greater than 0.15. These frames were immediately censored and not used in the estimation of FC.*
2. *We also censored any low-motion time points that were within two frames of any of the frames censored in step 1.*
3. *Following steps 1 and 2, we further censored any sequence of temporally contiguous low-motion frames that was shorter than five frames.*
4. *Lastly, we excluded a scan if more than 50% of its frames were flagged as high-motion following steps 1-3, i.e. were censored. If any scan from a given subject was removed, we removed that subject and all of their other scans from analysis.*

Given these criteria, we retained 70/95 HCP subjects. Of these remaining subjects, we retained, on average, $90.3 \pm 0.97\%$ of their frames. The frames that were retained had 0.07 ± 0.01 relative motion. In contrast, the censored frames had relative motion of 0.15 ± 0.03 .

Using these data, we fully replicated the results reported in the main text. Specifically, we made predictions of whole-brain FC using the 40 predictors from the 70 low-motion subjects. This resulted in a [40 predictor x 70 subject] matrix of R^2 values, which we transformed into a [2800 x 1] vector. We did the same using the data from the main text and computed the similarity (Pearson correlation) between these two vectors. We found a strong correspondence ($r = 0.997$; $p < 10^{-15}$). The findings held at the subject level in which we calculated the similarity of columns ($r = 0.999 \pm 1.3 \times 10^{-4}$). Next, we performed an analogous procedure at the regional level, resulting in a [400 x 40 predictor x 70 subject] matrix of R^2 values. Again, we assessed the similarity of the motion-corrected results with those reported in the main text by vectorizing each matrix, transforming them into vectors of [1120000 x 1], and computing their Pearson correlation. As before, we found an excellent correspondence ($r = 0.995$). We repeated this analysis at the subject level, transforming each of the 70 [400 x 40] matrices into [16000 x 1] vectors and computing separately for each subject, their similarity. Once again, we found a strong correspondence ($r = 0.9995 \pm 6.13 \times 10^{-4}$). Finally, we modeled FC using pairs of predictors with multilinear models, identified for each region the optimal pair (the one that explained the most variance), and summarized these results with a count matrix, as in the main text. We then computed the correlation of the motion corrected count matrix (its upper triangle) with the matrix reported in the original submission. We found a strong correlation of $r = 0.997$ ($p < 10^{-15}$).

We also performed a similar procedure using the NKI dataset. Specifically, we excluded subjects from analysis whose temporal signal to noise ratio (TSNR; mean BOLD signal across time divided by temporal standard deviation; Krüger et al 2001) was in the bottom P% or whose DVARS (derivative of RMS variance over voxels; Power et al 2012) was in the top Q%. Of the remaining subjects, we regressed out both variables from the regional R^2 values and analyzed the residuals.

Depending upon the values of P and Q, different subsets of subjects were excluded. Here, we systematically varied these parameters and report:

1. The number of retained subjects,
2. The similarity (correlation) of the residual R^2 vs. age correlation pattern with the pattern reported in the main text, and
3. The mean difference in correlation magnitude.
4. The mean difference in absolute correlation magnitude.

We show the results of this analysis in the below figure.

In general, we find broad correspondence between the results reported in the main text and the results at all combinations of parameters (see Fig. S13). Specifically, the similarity between correlation patterns never fell below a value of $r = 0.29$. However, this pattern was obtained using an unreasonable combination of parameters in which only 8 total subjects were retained. For parameter combinations that retained at least half of the full dataset (262 subjects), the minimum similarity was $r = 0.83$, with a mean \pm standard deviation of $r = 0.89 \pm 0.02$ and range of 0.83 to 0.92.

Over this same range the mean correlation magnitude decreased by -0.032 ± 0.005 while the mean absolute correlation increased by 0.035 ± 0.003 , suggesting that negative correlations are getting increasingly more negative.

Fig. S13. Effect on correlation of regional R^2 versus age of motion-correction applied to NKI dataset. In the main text we reported a correlation between regional R^2 and age using the NKI dataset. Here we assess the effect of motion-correction on these results. Broadly, this strategy involved excluding data from subjects in the bottom P% based on their temporal signal to noise ratio (TSNR; mean BOLD signal divided by temporal standard deviation (Kruger et al 2001), or was in the top Q% by DVARS (derivative of RMS over voxels; Power et al 2012). Of the remaining individuals, we regressed out both variables from their regional R^2 values and computed their correlation with age. Depending on the values of P and Q different subsets of subjects were retained. Here, we systematically varied these parameters and report: (a) The similarity of the R^2 versus age correlation pattern with the pattern reported in the main text. (b) The mean difference in regional correlation coefficients ($\frac{1}{400} \sum r_{motion\ corrected} - r_{original}$). (c) The mean difference in absolute regional correlation coefficients ($(\frac{1}{400} \sum |r_{motion\ corrected}| - |r_{original}|)$). (d) The total number of subjects retained. In each panel, we also plot parameter combinations that result in at least 263 subjects being retained (half of the original sample).

Collectively, the results of these two analyses support those reported in the main text. We now include the above figure (Fig. S13) in the Supplementary Material and have updated the text as well.

- *Floor effect of age-related R^2 differences.*

o The authors should address the possibility for a floor effect where regions with initially low R^2 (Figure 4d) have little room to exhibit age-related decreases in R^2 . A conjunction map of Fig. 4d and 4e would likely show areas of low R^2 showing little decrease or even increases, whereas areas with the highest R^2 to begin with has the greatest room to show ‘decreases’ with age. While it is likely that those areas truly exhibit greater age-related decreases in R^2 , the possibility of a floor effect or even regressing to the mean should be addressed.

This is an important question. We address the possibility of a floor effect using null models to generate null distributions and estimate the theoretical lower limits of regional R^2 values. If a floor effect exists for some regions, wherein their observed R^2 values cannot decrease further, we expect them to be consistent with the null.

For each subject, we predicted their regional FC patterns using predictors derived from a randomized SC matrix (the order of its rows and columns were permuted). We then calculated and retained, for each region, its maximum R^2 across any of the predictors. We repeated this procedure 100 times for each subject, generating subject-specific null distributions. We compared the observed R^2 values with this null distribution (z-score) and identified regions whose z-statistic could not be distinguished statistically from the null distribution ($p \geq 0.05$). We counted how frequently across subjects each of the 400 regions was part of this group, considering them to have already reached their floor R^2 .

We find that, on average, most regions exhibit R^2 values that are statistically greater than their theoretical floor ($96.8 \pm 1.5\%$). The specificity of regions that are approaching their R^2 floor is also poor; the region with the greatest frequency is consistent with the null distribution in 14.7% of subjects (Fig. S14a). Interestingly, and as the reviewer anticipated, the whole-brain pattern is similar to the correlation pattern reported in the main text ($\rho = 0.51$; $p < 10^{-15}$; Fig. S14b), opening up the possibility that some of the age-related differences may be attributable to a floor effect.

However, disentangling those two effects – the floor effect from true age-related differences – is challenging. Accordingly, we have included the above figure and now discuss the floor effect as a limitation and area for future investigation.

Fig. S14. In the main text we reported a correlation between regional R^2 and age using the NKI dataset. Here we assess the possibility that the spatial pattern of these correlations may be driven, in part, by a floor effect. To test for this effect, we use a permutation-based null model. Regions whose R^2 values cannot be statistically distinguished from the null distribution are especially susceptible to a floor effect. In more detail, this procedure entailed the following. For each subject, we predicted their regional FC patterns using predictors derived from a randomized SC matrix (the order of its rows and columns were permuted). We then calculated and retained, for each region, its maximum R^2 across any of the predictors. We repeated this procedure 100 times for each subject, generating subject-specific null distributions. We compared the observed R^2 values with this null distribution (z-score) and identified regions whose z-statistic could not be distinguished statistically from the null distribution ($p \geq 0.05$). We counted how frequently across subjects each of the 400 regions was part of this group, considering them to have already reached their floor R^2 . We find that, on average, most regions exhibit R^2 values that are statistically greater than their theoretical floor ($96.8 \pm 1.5\%$). The specificity of regions that are approaching their R^2 floor is also poor; the region with the greatest frequency is consistent with the null distribution in 14.7% of subjects (Fig. S14a). Interestingly, and as the reviewer anticipated, the whole-brain pattern is similar to the correlation pattern reported in the main text ($\rho = 0.51$; $p < 10^{-15}$; Fig. S14b), opening up the possibility that some of the age-related differences may be attributable to a floor effect.

consistent with the null distribution in 14.7% of subjects. Interestingly, and as the reviewer anticipated, the whole-brain pattern is similar to the correlation pattern reported in the main text ($\rho=0.51$; $p<10^{-15}$), opening up the possibility that some of the age-related differences may be attributable to a floor effect. (a) Fraction of subjects in which a region's R^2 is not distinguishable from the null distribution. We show these values projected onto the cortical surface. (b) Values from panel a with the regional R^2 versus age correlation coefficients. (c) Values from panel a grouped by system.

The revised text reads:

A final limitation concerns the possibility that the reported correlations between R^2 and age could be attributed to a floor effect. That is, any regions that start with low R^2 early in life necessarily have less room to decrease over the course of the lifespan, whereas regions that start with high levels of R^2 have more room to decrease. Although we show that the fraction of regions that may be susceptible to this effect is small, the spatial pattern is correlated with the reported R^2 versus age correlation map (see Fig. S14}). Disentangling true age-related differences from floor effects is challenging and may require the use of an unbounded measure other than R^2 for assessing structure-function correspondence. Future work should investigate these potential confounds in greater detail.

Minor:

- Age-related differences observed from a cross-sectional sample should be labeled as 'differences' instead of 'changes'; changes implies within-person changes, which requires longitudinal data.

We have changed all references to age-related changes or lifespan changes to differences.

- Yellow labels or colors are not very visible/readable. Please consider using something like "brown" (e.g., Fig 1c, 2a, 2g)

We appreciate the reviewer bringing this to our attention. We've changed the color to brown.

- Terms that are specific to other types of data (e.g., 'expression' from genes, 'fingerprinting') creates more confusion than actually help the reader understand the results.

We edited every instance where we used the word "express", "expression", "expressed", "overexpressed", "fingerprint", or "fingerprints".

- Due to how similar in setup Figure 1c and 2a are, their y-axis label should indicate that 2a is actually "Mean variance explained across subjects".

We have changed the axis label.

REVIEWER COMMENTS

Reviewer #1 (Remarks to the Author):

I highly appreciate the extensive set of new analyses the authors performed in response to the reviewer comments as well as the increased clarity of the manuscript. I have a few minor concerns left:

1. The additional analyses of the floor effect as a driver of regional differences in the age-R² correlations are an important complement to the paper. However, the results in figure 4G are currently described separately from the results in figure S14 while it seems to me that the associations that are described may be very similar. The floor effect may be a driver of the relationship between variance explained and its correlation with age. That is why I would suggest describing these findings in the same part of the results section.

2. All the figures shown in the manuscript should show results that indicate statistical significance. As they are presented now, the casual reader may be unaware that results are uncorrected. Unthresholded maps could be shown in the supplementary materials. For example, figure 4d-e should contain results after correlation for multiple comparisons, so it is clearer that some of these results are not statistically significant. Also in figure 4f, statistical significance should be indicated with asterisks. Figure 1c could be replaced by figure S1.

3. A related points is that the grey – colour gradient used in many figures results in a rather arbitrary and very stark threshold between the grey regions and the coloured ones. I initially mistook this threshold to reflect statistical significance and I think other readers might make the same mistake. That is why I would recommend using a different colour scale with a less stark transition.

4. I would suggest replacing figure 3D by figure S7. Since the results are nearly identical I do not see the added value of presenting both.

Reviewer #2 (Remarks to the Author):

The authors have answers all my and others comments in an excellent way. I recommend publication as it is. Thanks the authors for the the hard work in the revision.

Reviewer #3 (Remarks to the Author):

The authors have addressed multiple concerns raised in the previous review. Specifically, they have now clarified their choice of predictors in the introduction with supporting theory, included null models for global analysis, and explored how the 40 predictors perform using PCA in their supplemental analysis. However, there still remain issues regarding data-quality due to motion-related noise. While the authors

have done additional analyses to respond to the previous comments about movement, this limitation has not been adequately addressed in the NKI dataset which is the basis of the conclusions regarding age-related differences in the reported observations. This limitation is elaborated below, along with a number of other more minor comments.

Motion correction:

a. The authors performed motion correction using frame censoring on the HCP dataset, but not for the NKI dataset (which likely has greater motion contamination). The processing strategy in this latter dataset was to exclude subjects based on a threshold P% TSNR and Q% DVARS. While this latter procedure removes subjects that move more on average, the remaining subjects still retain motion contaminated frames which would bias the measurement of resting-state correlations. Given that the HCP data were frame censored, its not clear why this same approach was not conducted for the NKI dataset, especially given that the latter includes individuals that exhibit a higher number of large head movements during the scan session (i.e., older adults). Without adequate motion processing, the present conclusions regarding the lifespan variation are not supported.

b. In relation to the above, the authors compared results of the motion-corrected data to the original uncorrected analysis by comparing the vectorized matrix of the original analysis (i.e., from the main text) and the motion-corrected analysis. While the patterns appear comparable, this is an indirect way of demonstrating the results are robust to attempts to minimize sources of motion-related noise. This is particularly important for the lifespan dataset analysis using the NKI dataset. The authors should report the outcome statistics using the motion-corrected processing, both for HCP and NKI.

Minor comments:

- 2nd paragraph of pg. 3, the author states that “Across all factors, we found that the majority of variance explained can be attributed to one-step (direct) connections (Fig. 1c)”. I think they meant to refer to Fig. 1d.
- In the first paragraph of pg. 12, “Interestingly, while the regional patterns of structure- function coupling in NKI and HCP datasets were correlated, the magnitude of coupling was noticeably stronger.” Please clarify what is the magnitude of coupling noticeably stronger in.
- The authors should explain why age ‘bins’ are used in the lifespan analysis instead of using just age as a continuous variable.
- The authors clarified that their use of PCA in the main text was mainly to showcase the similarity between predictors in a lower dimensional space. Were the columns centered and/or normalized? Centering (and in some cases normalizing) is often the default for PCA functions, and could alter interpretation of the resultant output. Greater details should be provided to clarify what was done.
- In the paragraph before the Discussion section in pg. 8, “Heteromodal systems, like default mode and control networks, on the other hand, exhibit subtle reductions in coupling magnitude and, in some cases, even increase with age.” It is a leap to infer potential increases of an entire system’s effect when the mean $r(\text{age}, R^2)$ is below 0. The authors should revise to point out certain “regions” within the default mode and control networks may exhibit these patterns.

Dear Reviewers,

We wish to thank the editor and all reviewers for their thoughtful comments and suggestions. Our enclosed submission is significantly improved as a result. We provide, here, a short summary of the largest changes:

1. We updated the post-hoc motion censoring strategy for the NKI dataset so that it is consistent with the technique applied to the HCP data.
2. We updated all main text figures so that the reported results come from the motion-censored data rather than reporting uncensored data and, treating it as a benchmark, verifying its robustness with censored data.
3. We updated figure colormaps (Figure 2h) and added two additional supplementary figures; Figure S2 depicts the fraction of subjects for which regional structure-function coupling was statistically significant and Figure S10 shows age-frequency analysis without the binning procedure.
4. We have incorporated several panels from supplementary figures from the previous submission into the main text; those supplemental figures no longer appear. Specifically, we removed the figure showing thresholded correlation maps and the adjusted R^2 analysis.

Throughout this letter we refer to comments made by the reviewers in *italics*. Changes made to the text are shown in *blue italics* and indented. When appropriate, we highlight the section name where changes were implemented. In addition to this response letter, we include two versions of the revised manuscript: a “marked” version that tracks all of the changes made since the original submission and a “clean” version that includes all changes but without any highlights.

Farnaz Zamani Esfahlani

Rick Betzel

Reviewer #1 (Remarks to the Author):

I highly appreciate the extensive set of new analyses the authors performed in response to the reviewer comments as well as the increased clarity of the manuscript. I have a few minor concerns left:

1. The additional analyses of the floor effect as a driver of regional differences in the age-R² correlations are an important complement to the paper. However, the results in figure 4G are currently described separately from the results in figure S14 while it seems to me that the associations that are described may be very similar. The floor effect may be a driver of the relationship between variance explained and its correlation with age. That is why I would suggest describing these findings in the same part of the results section.

We appreciate this point and have moved the description of the floor analysis to the same section where we describe results in Figure 4g. As a result, Figure S14 is now numbered Figure S12. The revised text reads:

“An important concern related to age differences in structure-function coupling is the possibility of a floor effect. Namely, that the R² of some regions cannot exhibit significant decreases because it is already near its theoretical floor value. To assess the likelihood of such an effect occurring, we estimated floor values for each region and subject using a permutation-based null model (100 repetitions; see Fig. S12 for methodological details) and compared them against the observed R² values. In general, we found that most regions ($96.8 \pm 1.5\%$ were significantly greater than their theoretical floor) and that those regions nearing the floor did not overlap with the regions in which we reported strong age effects.”

2. All the figures shown in the manuscript should show results that indicate statistical significance. As they are presented now, the casual reader may be unaware that results are uncorrected. Unthresholded maps could be shown in the supplementary materials. For example, figure 4d-e should contain results after correlation for multiple comparisons, so it is clearer that some of these results are not statistically significant. Also in figure 4f, statistical significance should be indicated with asterisks. Figure 1c could be replaced by figure S1.

This is an important point. We have updated the following figures and panels to indicate statistical significance.

- Figure 4e – replaced with thresholded surface map.
- Figure 4f – added asterisks indicating system-level significance.
- Figure 1c – replaced with Figure S1.
- Figure 3e – added asterisks indicating system-level significance.
- Figure 2c and Figure 4d – these panels depict average R² and were left unchanged, as they represent complicated transformations of statistical models (means of the

maximum R^2 values across subjects and predictors, respectively). However, we have added an additional figure depicting the fraction of subjects for which each region was determined to be statistically significant (false discovery rate fixed for each subject independently at $q = 0.05$). Note that in almost every case, the models were statistically significant. It is straightforward to understand why this is. At the group level, the smallest R^2 values in the HCP dataset are close to 0.04. When transformed back to a correlation coefficient, this yields a value of $r = 0.2$. Given that the correlations were estimated using $N - 1$ samples and the monotonic relationship of r with p -values, correlations of this magnitude will tend to be statistically significant.

Fig. S2. Fraction of subjects with significant regional structure-function coupling. (a) Human Connectome Project dataset. (b) Nathan Kline Institute dataset. For both datasets, the false discovery rate was fixed at $q = 0.05$ and a separate adjusted p -value calculated for each subject.

3. A related point is that the grey – colour gradient used in many figures results in a rather arbitrary and very stark threshold between the grey regions and the coloured ones. I initially mistook this threshold to reflect statistical significance and I think other readers might make the same mistake. That is why I would recommend using a different colour scale with a less stark transition.

We appreciate this suggestion and apologize for any confusion. We presume that the reviewer is referring primarily to the colormap used in Figure 2h. In these panels, the colormaps go from a gray value to some color, where the color corresponds to a particular family of predictors, e.g. the red map represents the communicability measures. We agree with the reviewer that these maps can be improved, both visually and in terms of clarity.

To this end, we updated the colormaps and also added text in the figure caption, noting explicitly that the colors denote categories or families of predictors and not statistical significance.

In choosing new colormaps, we considered, specifically, the perceptual contrast of maps we currently use. In general, a concern about some colormaps is that they are perceptually imbalanced, such that variation in data values is not well captured by variation in the colormap itself. A classic example of a problematic colormap is the former “jet” map, which used to be

the default in MATLAB (see Kovesei, 2015 for a detailed explanation of the issues and this blogpost for Mathworks justification on why “jet” was replaced).

In examining our current colormaps using methods from Kovesei (2015), we found evidence of ranges of modest imbalance. These imbalances arose because of how we selected our base colors. Our current colormaps were generated using four ordered RGB triplets and interpolating between the extremes. Specifically, the four triplets were:

“light gray” → “dark gray” → “color” → “lighter version of same color”

As an example, consider the “dark blue” colormap, which corresponds to the family of predictors associated with mean first passage times (mfpt-wei and mfpt-bin). In the top panel of the figure below, we show the four base colors along with the interpolated colormap. Following Kovesei (2015), we superimposed sinusoids of varying amplitude over the colormap. Intuitively, if the colormap is perceptually balanced then the variation should be evident. Regions of the colormap where the sinusoidal behavior is not perceptually distinguishable correspond to perceptual discontinuities. Using this approach, we identified at least one region of perceptual discontinuity near the transition between “color” and the “lighter color” (a similar discontinuity was evident in the other colormaps). We highlight this in red in the below figure.

To resolve this issue, we created a new colormap that transitioned from gray to dark blue. We repeated the above analysis and found no clear evidence of discontinuities, suggesting that the new colormap represents an improvement. For each predictor (subpanels of Figure 2h), we

generated a new colormap following the above procedure. We show, below, the updated panel:

The revisions to the figure captions are shown below:

In Figure 1:

“Note that here and in all other figures, we adopt a consistent color scheme, wherein data associated with a given predictor, e.g. same measure but different parameters or weighted/binary versions of a measure, are depicted using variation of the same color. For example, communicability measures are red, matching index is yellow, and mean first passage time is dark blue.”

In Figure 2:

“Note that the base colors for each surface plot correspond to different predictors and not varying levels of consistency across subjects or statistical significance.”

4. I would suggest replacing figure 3D by figure S7. Since the results are nearly identical I do not see the added value of presenting both.

We agree and have done so. We have also removed the previously designated Figure S7 from the supplement. Finally, we have modified the text that had previously appeared in the caption for Figure S7 and created a new subsection in the Materials and Methods section, describing the procedure for calculating adjusted R^2 . The added text in both sections reads:

In the Results section:

“Here, when we calculate ΔR^2 , we adjust the multilinear model's R^2 to penalize for the addition of a second predictor (see **Materials and Methods** for details of the correction).”

In the Materials and Methods section:

“In the main text we reported the change in variance explained, ΔR^2 as a result of including two predictors. In general, we might expect increases in R^2 simply due to the inclusion of a second parameter. One strategy to account for this addition is to adjust the R^2 measure based on the number of predictors. Specifically, we calculated

$$R^2_{adjusted} = 1 - \frac{(1-R^2)(N-1)}{N-p-1}$$
 for each region for both the one- and two-term models.

Here, $N = 399$ is the number of samples ($N_{regions} - 1$ because we exclude self-connections) and p is the number of predictors in the model and is equal to $p = 2$ and $p = 3$ for the one- and two-predictor models (the additional parameter is from the intercept).”

Reviewer #2 (Remarks to the Author):

The authors have answers all my and others comments in an excellent way. I recommend publication as it is. Thanks the authors for the the hard work in the revision.

We thank the reviewer for the positive comments.

Reviewer #3 (Remarks to the Author):

The authors have addressed multiple concerns raised in the previous review. Specifically, they have now clarified their choice of predictors in the introduction with supporting theory, included null models for global analysis, and explored how the 40 predictors perform using PCA in their supplemental analysis. However, there still remain issues regarding data-quality due to motion-related noise. While the authors have done additional analyses to respond to the previous comments about movement, this limitation has not been adequately addressed in the NKI dataset which is the basis of the conclusions regarding age-related differences in the reported observations. This limitation is elaborated below, along with a number of other more minor comments.

Motion correction:

1. The authors performed motion correction using frame censoring on the HCP dataset, but not for the NKI dataset (which likely has greater motion contamination). The processing strategy in this latter dataset was to exclude subjects based on a threshold P% TSNR and Q% DVARS. While this latter procedure removes subjects that move more on average, the remaining subjects still retain motion contaminated frames which would bias the measurement of resting-state correlations. Given that the HCP data were frame censored, its not clear why this same approach was not conducted for the NKI dataset, especially given that the latter includes individuals that exhibit a higher number of large head movements during the scan session (i.e., older adults). Without adequate motion processing, the present conclusions regarding the lifespan variation are not supported.

We appreciate this point and have applied an identical motion correction procedure to the NKI data as we did with the HCP. Specifically, we identified motion-contaminated frames by first censoring frames whose relative motion was greater than 0.15. We further censored frames that were within 2 frames of those identified by the previous criterion. We then excluded any sequences of uncensored frames that were shorter than 5 contiguous frames. As a final control, we excluded any subject for whom 50% of their frames were censored.

The procedure resulted in an exclusion of 110 individuals. Of the remaining subjects, we retained, on average, $93.3\% \pm 10.0\%$ of their frames. The mean relative motion of those frames was 0.055 ± 0.014 . The motion of the excluded frames was 0.16 ± 0.06 .

We report these statistics in the main text:

“For the NKI dataset, we retained 432/542 subjects, keeping $93.3 \pm 10.0\%$ of frames, with a mean motion level of 0.06 ± 0.01 . The censored frames had a motion level of 0.16 ± 0.06 . We also calculated the number of NKI subjects within each of the 10 age bins that had been excluded and found that, as expected, frame censoring tended to exclude older subjects ($p = 0.47$; $p = 0.038$).”

To assess whether this censoring and exclusion procedure systematically impacted subjects of different age groups, we used the same bins as described in Figure 4c and calculated the fraction of subjects in that bin that had been excluded. As expected, we found that in bins corresponding to older individuals, a greater fraction of subjects had been excluded ($\rho(\text{age, fraction}) = 0.47, p = 0.038$).

As in the main text, we then constructed functional connectivity matrices using only low-motion frames and only for subjects meeting the inclusion criteria (50% of their frames were usable). We used the same structural predictors to model regional structure-function coupling.

Then, we followed the procedure described in the main text to link age and intelligence with coupling magnitude across regions. In general, we find an excellent correspondence with our originally reported results (similarity with respect to originally reported findings of $r = 0.82$ and $\rho = 0.81$; both $p < 10^{-15}$).

The reviewer's second comment is related to how we should report these and other motion-corrected findings. We continue our response below.

2. In relation to the above, the authors compared results of the motion-corrected data to the original uncorrected analysis by comparing the vectorized matrix of the original analysis (i.e., from the main text) and the motion-corrected analysis. While the patterns appear comparable, this is an indirect way of demonstrating the results are robust to attempts to minimize sources of motion-related noise. This is particularly important for the lifespan dataset analysis using the NKI dataset. The authors should report the outcome statistics using the motion-corrected processing, both for HCP and NKI.

We agree with the reviewer that reporting results/outcomes generated without motion censoring is not appropriate. To this end, we have done the following:

1. Materials and Methods: Updated the description of the frame censoring procedure.
2. Main text: updated all reported statistics so that come from the frame-censored data.
3. Main and supplemental figures: updated all figure panels so that they came from the frame-censored data. Removed Figure S13, which described the older method for correcting motion in NKI data.

Broadly, we find excellent correspondence between results with the original data and the data that underwent frame censoring and excluded high motion subjects. As noted about, the main finding – regional correlations of age with coupling magnitude – was largely preserved, as was the correlation with age and global coupling.

We note one important change. In our original submission, we used a binning procedure to link the frequency of different structural predictors with age. In addition to binary/weighted mean first passage time, we reported six other predictors that exhibited statistically significant

correlations. Here, after the motion-correction procedures, the only predictors that exhibited statistically significant age correlations were the two mean first passage time measures (mfpt-wei and mfpt-bin). These measures also were the only measures that exhibited significant age correlations at the subject level without the binning procedure.

In summary, we appreciate that the reviewer requested this additional procedure for motion correction. Not only does it maintain consistency with the procedure applied to the HCP dataset (previously the two motion correction methods were different), but it helped us reduce the likelihood of spurious age effects, vis-à-vis the age-predictor frequency correlations.

Because the changes to the main text and figures involve mostly updating statistics and figure panels to be consistent with the motion-corrected data, we do not report them all here.

Minor comments:

1. 2nd paragraph of pg. 3, the author states that “Across all factors, we found that the majority of variance explained can be attributed to one-step (direct) connections (Fig. 1c)”. I think they meant to refer to Fig. 1d.

We thank the reviewer for pointing this out. We have updated the error.

5. In the first paragraph of pg. 12, “Interestingly, while the regional patterns of structure-function coupling in NKI and HCP datasets were correlated, the magnitude of coupling was noticeably stronger.” Please clarify what is the magnitude of coupling noticeably stronger in.

We appreciate the opportunity to respond. Our claim was made without statistical support and we have removed the statement that one or the other datasets exhibited greater/weaker values of local structure-function coupling. The revised text now simply notes that the coupling patterns are not identical between datasets:

“Interestingly, while the regional patterns of structure-function coupling in NKI and HCP datasets were correlated, the coupling patterns were not identical.”

3. The authors should explain why age ‘bins’ are used in the lifespan analysis instead of using just age as a continuous variable.

We appreciate the opportunity to clarify. Here, the binning procedure was used to amplify signal across noisy individual observations. However, we also performed the same analysis using age as a continuous variable, as suggested by the reviewer. In general, we found

corroborative evidence using this approach, with both mean first passage time measures (mfpt-wei and mfpt-bin) exhibiting statistically significant correlations with age: $r_{mfpt_wei} = -0.32$ ($p = 9.8 \times 10^{-13}$) and $r_{mfpt_bin} = -0.22$ ($p = 7.7 \times 10^{-7}$). Here, we adjusted the critical p-value by fixing the false discovery rate at $q = 0.05$ ($p_{adj} = 0.0011$). No other predictors were statistically significant. We now include this analysis and corresponding scatterplots in the supplement.

Figure S10. Correlation of predictor frequency with age as a continuous variable. In the main text we binned subjects into groups based on their age and calculated the correlation frequency of each predictor versus age. Here, we repeat this analysis but without the binning procedure. We found that only weighted and binary mean first passage time passed multiple comparison corrections ($r = -0.32$ and $r = -0.22$; $p = 9.8 \times 10^{-13}$ and $p = 7.7 \times 10^{-7}$; false discovery rate fixed at $q = 0.05$; adjusted critical value of $p_{adj} = 0.0011$).

The revised text reads:

“We then grouped subjects into percentile-based age bins (10 bins in the main text; see Fig. S9 for reproducibility of results with different numbers of bins and Fig. S10 for results in which age was treated as a continuous variable and without the binning procedure), and found that ...”

4. The authors clarified that their use of PCA in the main text was mainly to showcase the similarity between predictors in a lower dimensional space. Were the columns centered and/or normalized? Centering (and in some cases normalizing) is often the default for PCA functions, and could alter interpretation of the resultant output. Greater details should be provided to clarify what was done.

We apologize for the omission. Here, we did not center or z-score columns. We now note this. The revised text reads:

In Figure 2’s caption:

“Note that here the principal component analysis was carried out without z-scoring or centering the columns of the correlation matrix.”

In Figure 3’s caption:

“As in Fig. 2b, we did not center or z-score columns as part of the principal component analysis.”

5. In the paragraph before the Discussion section in pg. 8, “Heteromodal systems, like default mode and control networks, on the other hand, exhibit subtle reductions in coupling magnitude and, in some cases, even increase with age.” It is a leap to infer potential increases of an entire system’s effect when the mean $r(\text{age}, R^2)$ is below 0. The authors should revise to point out certain “regions” within the default mode and control networks may exhibit these patterns.

We have updated the text in that section. It now reads:

“Collectively, these results suggest that the interrelationship of structural and functional connectivity covaries weakens with age. Notably, the areas that exhibit the greatest reductions fall within sensorimotor systems, which are among those with the strongest coupling to begin with, and include regions in striate and extrastriate cortex, as well as primary motor cortex and superior parietal lobule. Areas within heteromodal systems, like default mode and control networks, on the other hand, exhibit subtle reductions in coupling magnitude and, in some cases, even increase with age for instance, both dorsal and ventral prefrontal cortices along with temporal pole). Our findings point to heterogeneous differences in the complex relationship between the brain’s physical wiring and its intrinsic functional organization.”

REVIEWER COMMENTS

Reviewer #3 (Remarks to the Author):

The application of more rigorous noise-mitigation processing techniques in the NKI data is appreciated. The authors note that one consequence of this procedure is that participants that are older have fewer 'clean' frames, given that they move more. It is possible that some of the age-related differences in variance explained (global and regional structure-function correspondence) relate to the number of frames included in the analysis. Other measures have been regressed out (sex, time of visit, global network properties), and it would be important to also demonstrate that the observed age-related differences throughout figure 4 are not due to differences in data quantity used to calculate f_c .

I also think that the use of age bins continues to exaggerate the effects of age ($r > 0.9$ using age bins vs. $r > 0.3$ using continuous age). Given the emphasis of age as a continuous variable which is appropriate, it is important to retain the variance in continuous age estimates in the figure and replace 4c with versions of s10a/b. The current Figure 4C will likely mislead readers who might miss that those are binned age, and falsely attribute a much higher effect size of age.

Dear Reviewers,

We wish to thank the editor and all reviewers for their thoughtful comments and suggestions. Our enclosed submission is significantly improved as a result. Our edits consisted of:

1. Further refining the coupling-vs-age correlations by regressing out two motion-related nuisance variables (number of clean frames used to estimate FC and the mean framewise displacement of those frames) and ...
2. Replacing the binned correlations previously shown in Fig. 4c with plots where age is treated as a continuous variable.

Throughout this letter we refer to comments made by the reviewers in *italics*. Changes made to the text are shown in *blue italics* and indented. When appropriate, we highlight the section name where changes were implemented. In addition to this response letter, we include two versions of the revised manuscript: a “marked” version that tracks all of the changes made since the original submission and a “clean” version that includes all changes but without any highlights.

Farnaz Zamani Esfahlani

Rick Betzel

Reviewer #3 (Remarks to the Author):

The application of more rigorous noise-mitigation processing techniques in the NKI data is appreciated. The authors note that one consequence of this procedure is that participants that are older have fewer 'clean' frames, given that they move more. It is possible that some of the age-related differences in variance explained (global and regional structure-function correspondence) relate to the number of frames included in the analysis. Other measures have been regressed out (sex, time of visit, global network properties), and it would be important to also demonstrate that the observed age-related differences throughout figure 4 are not due to differences in data quantity use to calculate fc .

This is a good point – the number of frames we retained differed across individuals and was correlated with age, and we should try and correct for this effect.

Our strategy for doing so was to include number of censored frames (along with the framewise displacement of the frames that were retained) as covariates and to compute the correlation of the residuals with age.

This procedure was used to obtain new results for Figure 4a. The previously reported correlation of $r = -0.21$ is now reduced to $r = -0.11$ ($p = 0.02$).

We also corrected the regional correlation values reported in Figures 4e, 4f, and 4g. Including number of frames and framewise displacement of the retained frames produced correlation values that, overall, were slightly weaker than what we had originally reported (paired sample t -test; $t(399) = -12.73$, $p < 10^{-15}$), but with an overall similar pattern across cortical regions ($r = 0.84$). After correcting for multiple comparisons (FDR fixed at $q = 0.05$; $p_{adj} = 0.0082$), we find 16.25% of all regions (65) exhibited significant effects, most of which which were negative (coupling decreased with age). Previously, 43% (172) of all regions exhibited significant effects.

We have replaced Figures 4a, 4f, 4g, and 4h with the above results using the correlation coefficients obtained using this procedure. The updated figure is shown above.

Additionally, we have updated the main text where we describe the covariates and results of the impacted analyses. The revised text reads:

First, we assessed how in global structure-function correspondence varied with age. To do this, we calculated the maximum R^2 for each participant across all predictors. Then, to rule out the possibility that inter-individual differences in variance explained is related to differences in sex, time of visit (for data acquisition), data quality measures like the number of uncensored frames and the framewise displacement associated with those frames, or global network properties like total weight and binary density, we regressed these values out of each subjects' R^2 value. The residuals obtained following this procedure are, by definition, orthogonal to those nuisance variables. Finally, we calculated the linear correlation of these residuals with subjects' ages and observed that the two were significantly associated with one another ($r = -0.11$; $p = 0.02$); Fig. 4a, suggesting that the magnitude of structure-function correspondence decreases monotonically with age. Globally, the most common optimal predictors of FC were Euclidean distance (63% of participants) and weighted mean first passage time (33%) (Fig. 4b).

The previous analysis focused on global coupling between structure and function. Next, we investigated age-related differences in structure-function coupling at a local (regional) level. As with the global analysis, we regressed out the effect of sex, time of visit, the number of frames used to estimate FC, and the mean framewise displacement of "clean" frames along with global network properties. Because we were examining effects at the level of individual nodes, we also regressed out the effect of nodes' binary and weighted degrees. First, we asked whether the prevalence of certain predictors varied with age. For each region, we identified the predictor that best explained its regional pattern of FC and, for each subject, calculated the fraction of regions best explained by each factor. Then, we calculated the correlation of frequencies with biological age (Fig. 4c). We found that only weighted and binary mean first passage time were significantly correlated with age ($r = -0.32$ and $r = -0.22$; $p = 9.8 \times 10^{-13}$ and $p = 7.7 \times 10^{-7}$; false discovery rate fixed at $q = 0.05$; adjusted critical value of $p_{adj} = 0.0011$). We also repeated this analysis after binning subjects according to their ages and found similar results irrespective of bin size (see Fig. S9).

We also updated the figure captions for Fig. 4 and Fig. S9. The caption for Fig. 4 includes the text:

... (b) Distribution of optimal predictors for each subject across the lifespan. (c) Correlations between age and the frequency with this weighted and binary mean first passage time were the optimal predictor of regions' FC patterns. (d) Whole-brain pattern of variance explained. (e) Correlation of variance explained with age projected onto the cortical surface. ...

The caption for Fig. S9 now reads:

Effect of bin size on the correlation of predictor frequency with age. In the main text we reported a correlation between the frequency with which given predictors are optimal for a given region and age. In that analysis, each data point corresponded to an individual. Here, we partitioned subjects into age bins of varying size, calculated the mean age and frequency of each bin, and computed the correlation between these bin-level estimates. Here, we report results from this analysis. (a) With a bin size of 10, we find that only the weighted and binary mean first passage times were correlated with age. (b) In general, we show that these correlations persist across a broad range of age bins (from 5 to 50 in increments of 5). We also find that other measures are correlated at different bin sizes, but not consistently.

I also think that the use of age bins continues to exaggerate the effects of age ($r > 0.9$ using age bins vs. $r > 0.3$ using continuous age). Given the emphasis of age as a continuous variable which is appropriate, it is important to retain the variance in continuous age estimates in the figure and replace 4c with versions of s10a/b. The current Figure 4C will likely mislead readers who might missed that those are binned age, and falsely attribute a much higher effect size of age.

This is another important point and we apologize for any figures that might appear misleading. Following the reviewer's suggestion and, as we note above, we have replaced the binned figures (Figure 4c) with the continuous values. The binned panels now appear in Figure S9 (they are labeled panel a; the previous panel is now labeled b).

We have moved the text describing the binning procedure from the main text into the caption of Figure S9 (see above). We have also removed the Figure previously labeled S10, which included the correlations reported with age.

REVIEWERS' COMMENTS

Reviewer #3 (Remarks to the Author):

The authors have addressed my remaining concerns and I appreciate their continued responsiveness.